# Straight to Zero: Why Linearly Decaying the Learning Rate to Zero Works Best for LLMs

**Shane Bergsma, Nolan Dey, Gurpreet Gosal, Gavia Gray, Daria Soboleva & Joel Hestness**
Cerebras Systems
{shane.bergsma,joel}@cerebras.net

## Abstract

LLMs are commonly trained with a learning rate (LR) warmup, followed by cosine decay to 10% of the maximum ($10\times$ decay). In a large-scale empirical study, we show that under an optimal peak LR, a simple linear decay-to-zero (D2Z) schedule consistently outperforms other schedules when training at compute-optimal dataset sizes. D2Z is superior across a range of model sizes, batch sizes, datasets, and vocabularies. Benefits increase as dataset size increases. Leveraging a novel interpretation of AdamW as an exponential moving average of weight updates, we show how linear D2Z optimally balances the demands of early training (moving away from initial conditions) and late training (averaging over more updates in order to mitigate gradient noise). In experiments, a 610M-parameter model trained for 80 tokens-per-parameter (TPP) using D2Z achieves *lower* loss than when trained for 200 TPP using $10\times$ decay, corresponding to an astonishing 60% compute savings. Models such as Llama2-7B, trained for 286 TPP with $10\times$ decay, could likely have saved a majority of compute by training with D2Z.

## 1 Introduction

Learning rate (LR) schedules play an important role in training large language models. The original Transformers paper proposed a brief LR warmup followed by decay proportional to the inverse square root of the step number (Vaswani et al., 2017). This schedule has the advantage of not requiring prior specification of the total training steps. However, cooling down to a specific minimum LR is acknowledged to be "preferable when one knows the training duration in advance" (Zhai et al., 2022) as it produces "slightly better results" (Raffel et al., 2020). In this paper, our main focus is finding LR schedules that achieve the minimum loss given a pre-specified number of training tokens.

The "predominant choice" (Hu et al., 2024) in such training — the "de-facto standard" (Hägele et al., 2024) — is warmup followed by cosine decay to 10% of the maximum LR, an approach used in GPT3 (Brown et al., 2020), Gopher (Rae et al., 2022), Chinchilla (Hoffmann et al., 2022), BLOOM (Scao et al., 2023), Llama (Touvron et al., 2023a), Llama2 (Touvron et al., 2023b), Falcon (Almazrouei et al., 2023), Pythia (Biderman et al., 2023), etc. It is used "following Hoffmann et al." (Muennighoff et al., 2023), and is the default schedule in LLM codebases (Karpathy, 2024).

We present a large-scale empirical study to determine which schedules work best in which situations, and why. We focus on both *compute-efficient* and *over-trained* models. According to Chinchilla scaling laws (Hoffmann et al., 2022), the fewest FLOPs to achieve a given loss is obtained when models are trained for around 20 tokens-per-parameter (TPP). It is also common to train for more than 20 TPP because smaller, over-trained models are cheaper to serve (Touvron et al., 2023a). Our experiments (across various model scales, vocabulary sizes, and dataset sources) reveal a consistent outcome: when all schedules use their optimal peak LR, linear decay-to-zero (D2Z) works best at compute-optimal TPP. Moreover, the relative benefit of D2Z over $10\times$ (in terms of training, validation and downstream loss) *increases* with TPP. Compute savings from D2Z can be substantial for over-trained models (Figure 1).

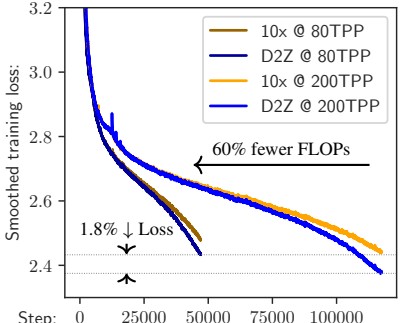

Figure 1: A 610M model trained for 80 TPP with *Linear*-D2Z has better train (and *validation*) loss than when trained for 200 TPP with *Linear*-$10\times$.

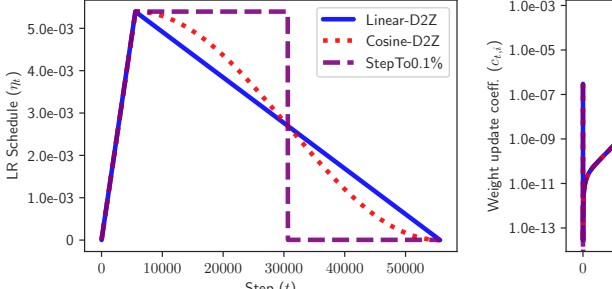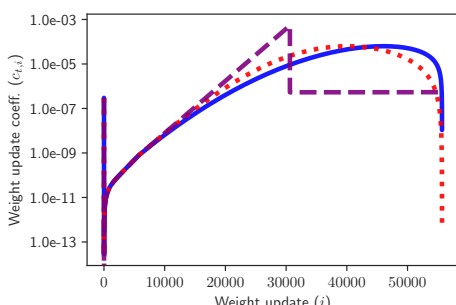

Figure 2: **LR schedules and their update-combination duals**: Each LR schedule, $\eta_t$ (left) and weight decay, $\lambda$, implies a weighted combination of weight *updates*, with combination coefficients $c_{t,i}$ (right, log-scale) giving the contribution of $i$th update to parameters $\theta_t$ at step $t$ (111M scale, coefficients at final step). The more sudden the drop in LR, the less emphasis on valuable later updates, perhaps explaining why *Step* underperforms *Cosine* and *Cosine* underperforms *Linear* decay.

LLMs such as Llama2-7B (Touvron et al., 2023b), trained for 286 TPP at $10\times$ decay, could likely have saved most of their training compute by switching to D2Z.

To explain the success of *Linear*-D2Z, we build on recent work on the related topic of *weight decay* (Andriushchenko et al., 2023; Wang & Aitchison, 2024). First, decaying *to zero* helps because compute-efficient training includes a long phase in which gradient noise is the key factor slowing the loss reduction; with higher noise resulting from higher TPP *or* smaller batches, a vanishing LR works best. Second, we show that approaching zero *linearly* is beneficial via a novel interpretation of AdamW (Loshchilov & Hutter, 2017), the primary optimizer in LLM pre-training. With AdamW, weights generated at each step are implicitly a weighted combination of weight *updates*. The combination coefficients depend on the learning rate schedule and weight decay settings (Section 3.1). Analyzing this *dual* of the LR schedule, we observe linear decay to produce a favorable combination of prior weight updates (Figure 2). When LR drops abruptly, e.g., via step-decay or, to a lesser extent, cosine decay, later updates receive less emphasis, leading to worse model quality. The EMA dual perspective also reveals the implicit schedule-awareness of recent "schedule-free" approaches such as *Warmup-Stable-Decay* (WSD) (Hu et al., 2024; Bi et al., 2024; Hägele et al., 2024).

Our empirical study encompasses hundreds of models trained over a grid of schedules and peak LRs, with scales ranging from 111M to 1.7B parameters, and datasets up to 137B tokens. We also compare LR schedules over the cross product of weight decay versus peak LR, and peak LR versus batch size. We confirm a slight, consistent advantage of linear D2Z over cosine D2Z at 20 TPP. We also demonstrate that linear D2Z improves over continuous schedules such as WSD. Further, when dataset or batch size changes, optimal peak LRs are much more *stable* when using D2Z compared to using lesser decay. This latter finding exposes LR decay ratio as an important confounder in prior work studying optimal hyperparameter transfer with $\mu$P (Yang & Hu, 2020; Yang et al., 2021).

## 2  BACKGROUND AND RELATED WORK

### 2.1  LEARNING RATE SCHEDULES

LR schedules have a long history in stochastic optimization, and are motivated by convergence bounds for stochastic gradient methods (Moulines & Bach, 2011; Bottou et al., 2018). For example, following Andriushchenko et al. (2023), consider SGD for a convex loss parameterized by $\theta$: with a constant LR $\eta$, the gap between the optimum and current loss at step $t$ can be bounded by:

$$\mathbb{E}[\mathcal{L}(\theta_t) - \mathcal{L}(\theta_*)] \leq (1 - \eta\mu)^t ||\theta_0 - \theta_*||^2 + \eta\sigma^2 \tag{1}$$

where $\theta_0$ are initial parameters, $\theta_*$ is the loss minimizer, $\sigma^2$ is a bound on the variance of gradient noise, and $\mu$ is a measure of the objective's curvature (see also Bottou et al. (2018, Theorem 4.6)). A larger LR can decrease dependence on initial conditions (the *bias* term), but also increase the effect of gradient noise (the *variance* term). As training progresses, bias decreases exponentially in $t$, and the relative importance of variance increases. Bottou et al. (2018) note this motivates a strategy where LR is high initially (to mitigate bias) and lowered later (to minimize variance).

In practice, when and how to reduce the LR is rarely informed by ML theory. Many LLMs simply follow the 10x cosine schedule, which is noted to work slightly better than cosine D2Z in the influential work of Hoffmann et al. (2022). It is also well established that using a portion of a longer (or extending a shorter) schedule is suboptimal compared to using a schedule that reaches its minimum only at the final training step (Li et al., 2019; Hoffmann et al., 2022; Hu et al., 2024; Hägele et al., 2024). Linear decay after warmup (Howard & Ruder, 2018) has been used in LLMs, typically also to 10% of the peak (Henighan et al., 2020; Dey et al., 2023; Sengupta et al., 2023). Dropping LR at specific milestones (*step decay*) is popular in vision models (He et al., 2016; Zagoruyko & Komodakis, 2016; Li et al., 2020), but has also been used in LLMs (Bi et al., 2024).

Kaplan et al. (2020) compared various decay functions and concluded the specific schedule was unimportant given a high enough average LR, although decaying to zero "gives a fixed improvement close to the end of training." Few papers explicitly compare different LR schedules for large-scale training, and when comparisons are made (Shallue et al., 2019; Kaplan et al., 2020; Schmidt et al., 2021; Hoffmann et al., 2022; Yang et al., 2021), they are not the primary focus. So, while some insights are gained, "comprehensive study" is usually regarded as "out of scope" (Aleph Alpha, 2024). One exception is Defazio et al. (2023), who found linear equals or outperforms other common schedules, including cosine, across a range of problems, including LLM training. In addition, Defazio et al. develop convergence bounds that theoretically motivate linear as the optimal schedule. Unlike our work, they do not evaluate LLMs with different peak LRs or decay ratios.

Seeking to measure model quality at different training durations without having to re-train separate models from scratch (Zhai et al., 2022; Hägele et al., 2024), researchers have adopted various *continuous* schedules, such as constant, cyclic (Smith, 2017), etc. Following optimization theory (Moulines & Bach, 2011; Defazio et al., 2024) weight averaging provides an alternative to decay (Sandler et al., 2023; Sanyal et al., 2023; Busbridge et al., 2024), although it is typically not as effective (Hägele et al., 2024), and moreover may have hyperparameters that implicitly depend on training duration (Defazio et al., 2024). *Warmup-Stable-Decay* (WSD) approaches have also been used in LLM training (Hu et al., 2024; Shen et al., 2024a; Ibrahim et al., 2024; Hägele et al., 2024). These methods train at a constant LR, but decay from a checkpoint in a separate process when an intermediate model is needed. We show that the optimal constant LR of these methods implicitly depends on training duration, rendering them not truly *schedule-free*.

## 2.2 Maximal update parameterization ($\mu$P)

It is common to scale initial weights of neural networks such that activations have unit variance (Glorot & Bengio, 2010; He et al., 2015). Yet weights can become unstable after a few steps of updates, if layer-wise LRs are imbalanced (Yang et al., 2021). In contrast, $\mu$P (Yang & Hu, 2020; Dey et al., 2024a) prescribes a *re-parameterization* of initial weight variances and LRs – essentially, rules for scaling these values as model width (i.e., $d_{model}$) scales – so that activations and updates remain stable. $\mu$P also stabilizes embeddings, layer norms, and self-attention in Transformers. $\mu$P is seeing growing application in LLMs (Dey et al., 2023; Shen et al., 2024b; Hu et al., 2024), where it acts to stabilize training and to enable transfer of optimal hyperparameters (HPs) across model scales.

With $\mu$P, base HPs can be tuned on a small *proxy*, or *base*, model, and transferred to larger models. Given the width of the proxy model, $d_p$, and width of the target, $d_t$, $\mu$P prescribes scaling factors to apply to HPs. The base LR $\tilde{\eta}$ is scaled down to $\eta = \rho\tilde{\eta}$, where $\rho = d_p/d_t$. In terms of LR schedules, the base LR $\tilde{\eta}_t$ is scaled at every step to provide $\eta_t$. $\mu$P is convenient in our study as we can sweep the same *base* peak LRs, $\tilde{\eta}$, at each model size, and observe trends that are scale-invariant.

## 2.3 AdamW weights as an exponentially-weighted moving average (EMA)

An AdamW update at a single training step, $t$, can be expressed as:

$$\theta_t = (1 - \eta\lambda)\theta_{t-1} - \eta\frac{\hat{m}_t}{\sqrt{\hat{v}_t} + \epsilon} \tag{2}$$

where $\eta$ is the learning rate, $\hat{m}_t$ and $\hat{v}_t$ are (bias-corrected) running averages of the gradient and the squared gradient, respectively, and $\epsilon$ is a small constant added to prevent division by zero. The weight decay value, $\lambda$, is typically set to 0.1 in LLM training (Brown et al., 2020; Hoffmann et al., 2022; Almazrouei et al., 2023; Aleph Alpha, 2024).

While the running averages of $m$ and $v$ in Adam (Kingma & Ba, 2014) and AdamW are exponentially-weighted moving averages (EMAs), Wang & Aitchison (2024) recently showed that the *weights* generated by AdamW can also be understood as an EMA — of the weight *updates*. That is, a standard EMA, $y_t$, for a time-varying quantity, $x_t$, can be written as:

$$y_t = (1 - \alpha)y_{t-1} + \alpha x_t \tag{3}$$

where $\alpha$ is the *smoothing parameter*. AdamW in Equation (2) can be seen as an EMA by letting:

$$y_t = \theta_t, \alpha = \eta\lambda, \text{ and } x_t = -\frac{1}{\lambda}\frac{\hat{m}_t}{\sqrt{\hat{v}_t} + \epsilon} \tag{4}$$

Wang & Aitchison note the quantity $\tau = 1/\alpha$, i.e., $1/(\eta\lambda)$, provides a rough *timescale* over which updates are averaged. Weight decay can therefore be used to control the effective window over which updates are combined (smaller values of $\lambda$ increase $\tau$, increasing the role of earlier updates in $\theta_t$). This perspective also motivates dynamic LR schedules. Initial $\tau$ should be small (high $\eta_t$), to forget early updates, "while the final timescale is around the total number of epochs [low $\eta_t$], to ensure averaging over all datapoints." In fact, we will show that contribution of weight updates, $x_0, x_1, \ldots x_t$, to $\theta_t$ at a particular step, $t$, cannot be determined by the *instantaneous* value of $\eta_t\lambda$. Rather, the contribution of any $x_i$ to $\theta_t$ requires looking at the full LR schedule *holistically* (Section 3.1).

Wang & Aitchison also motivate scaling rules for optimal $\lambda$ as model and dataset size vary. If dataset size increases by a factor of $K$ (i.e., $K\times$ as many steps), the EMA perspective recommends scaling $\lambda$ by $1/K$ in order to expand $\tau$ proportional to $K$. Moreover, with $\mu$P, if model size increases and LR is scaled by $\rho$ (Section 2.2), the EMA perspective motivates scaling $\lambda$ by $1/\rho$ to keep $\tau$ constant.[1]

## 3 METHODS, CONCEPTUAL FOUNDATIONS, AND HYPOTHESES

We now present an extended EMA perspective on AdamW that accounts for time-varying LRs. We then introduce our conceptual model of LLM training, and connect it to the EMA perspective. Finally, we outline the specific testable hypotheses that follow from our conceptual model.

### 3.1 ADAMW AS CONVEX COMBINATION OF WEIGHT UPDATES, DRIVEN BY LR SCHEDULE

Consider a generic moving average, but now with time-varying smoothing, $\alpha_t$, i.e., $y_t = (1 - \alpha_t)y_{t-1} + \alpha_t x_t$. If we let $\alpha_1 = 1$ (so that $y_1 = x_1$), we can express $y_t$ in terms of all inputs $x_t$:

$$
\begin{aligned}
y_1 &= \alpha_1 x_1, \\
y_2 &= (1 - \alpha_2)\alpha_1 x_1 + \alpha_2 x_2, \cdots \\
y_t &= \sum_{i=1}^{t} \left( \prod_{j=i+1}^{t} (1 - \alpha_j) \right) \alpha_i x_i \\
&= \sum_{i=1}^{t} c_{t,i} x_i
\end{aligned}
\tag{5}
$$

Where $c_{t,i} = \left( \prod_{j=i+1}^{t}(1 - \alpha_j) \right) \alpha_i$. It can be shown $\forall t, \sum_i c_{t,i} = 1$, and hence, as in a standard EMA, each $y_t$ is a *convex combination* of inputs $x_1 \ldots x_t$ (all coefficients are non-negative and sum to 1). However, with time-varying $\alpha_t$, we are not restricted to exponentially-decreasing $c_{t,i}$ as $i$ decreases. Indeed, for *any* convex combination of inputs, there is a corresponding smoothing schedule that generates the combination via the EMA. In terms of LR schedules for AdamW training, $y_t = \theta_t$, while $\alpha_t = \eta_t\lambda$ becomes the smoothing parameter at step $t$ (cf. Equations (3) and (4)).[2]

### 3.2 CONCEPTUAL MODEL: BIAS AND VARIANCE IN LLM TRAINING

Following Andriushchenko et al. (2023), our main conceptual premise is that in LLM training, there is an initial training phase focused on movement from initial conditions (bias reduction), followed by

---

[1] While the EMA perspective does not account for the updates $x_t$ themselves depending on parameters $y_{t-1}$, we find it a useful part of our *conceptual* model of training, in that it helps predict experimental results.

[2] When using a $\mu$P LR scaling factor, $\rho$ (Section 2.2), the smoothing parameter is $\alpha = \eta\lambda = \rho\tilde{\eta}\lambda$.

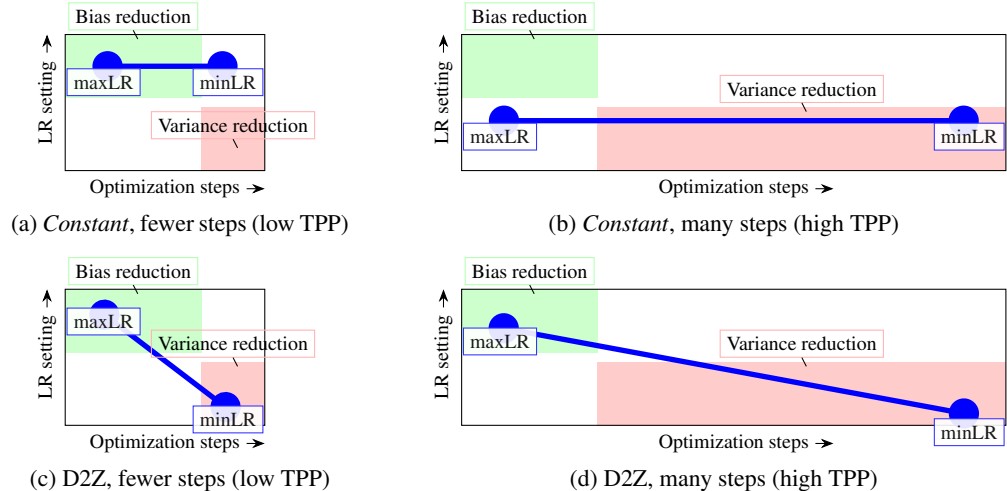

Figure 3: **Bias & variance in LLM pre-training**: as training duration increases (higher TPP), the importance of variance reduction — and having a lower LR — increases. With no decay (*Constant*, 3a, 3b), the optimal peak LR must drop significantly lower, neglecting bias reduction. With D2Z (3c, 3d), periods of bias and variance reduction can both be enjoyed without large shifts in peak LR.

a later phase bottlenecked by gradient variance. Furthermore, analogous to prior work optimizing the convex loss gap via SGD (Equation (1)), we argue the primary beneficial mechanism of LR decay in LLM training is to reduce gradient noise during later stages of training, while simultaneously minimizing the impact on bias reduction. The larger the dataset, the longer the proportion of training bottlenecked by gradient noise, and the greater the benefit from D2Z (Figure 3).

**Variance reduction** Per-step gradient variance is known to increase over the course of training (McCandlish et al., 2018). Very recently, Zhang et al. (2024) showed that for an LLM of a fixed size, training with larger datasets corresponds to larger *critical batch sizes*, which directly relates to larger (aggregate) gradient variance via the gradient noise scale (McCandlish et al., 2018).

In the EMA perspective, parameters $\theta_t$ are a combination of prior weight updates (indexed by $i$). The more variance at step $t$, the more updates that should be combined in order to reduce it. Here variance is reduced by combining updates *across* steps, rather than increasing batch size at a *specific* step. Now, with a constant LR, update coefficient $c_{t,i}$ decreases exponentially in $(t-i)$: the further back the update from the current step, the less it contributes. But if LR $\eta_i$ *decreases* over steps, each later coefficient $c_{t,m}$ will be smaller due to a lower $\alpha_m = \eta_m\lambda$. Yet since coefficients sum to one, *earlier* coefficients will correspondingly *increase*: coefficients are basically flattened and updates contribute more evenly across steps. The more we decay, the more updates we average over, and the more variance is reduced.

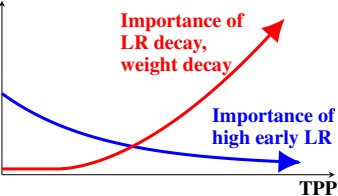

Figure 4: **HP influence vs. TPP**: Higher TPP means higher gradient noise; LR decay & weight decay settings thus increase in importance with TPP.

**Bias reduction** Higher decay is preferable to reducing *peak* LR because we must also minimize *bias*, i.e., the contribution of the initial random weights. In the EMA perspective, the contribution of these weights to $\theta_t$ is $c_{t,1} = \prod_{j=2}^{t}(1 - \alpha_j)$. For a decaying LR, and where $\alpha_j = \eta_j\lambda \ll 1$, coefficient $c_{t,1}$ can be approximated (with equality when LR is constant, Appendix B.1):

$$c_{t,1} \approx (1 - \bar{\alpha})^{t-1} \tag{6}$$

where $\bar{\alpha}$ is the average $\alpha_j$ over the schedule. Exactly as in Equation (1), bias therefore decreases exponentially in the *absolute* number of steps, $t$, with a rate of decrease depending on the LR. Crucially, this means that as we train for more total steps (i.e., a higher tokens-per-parameter; TPP), there is a decrease in the *fraction* of steps required for $c_{t,1}$ to become negligible. At higher TPP, bias reduction becomes relatively less important than variance reduction (Figure 4).

### 3.3 EXPERIMENTAL HYPOTHESES

**Hypothesis 1**: *As TPP increases, the relative benefit of D2Z over 10× decay will increase.*

This hypothesis follows from the premise that gradient variance plays a larger role at higher TPP, and greater LR decay (as in D2Z) allows for more updates to be averaged, and thus greater variance reduction. A related hypothesis is that if we increase the batch size at each step, gradient variance will decrease, and so the benefits of D2Z over 10× decay should *diminish*. Note our conceptual framework does not say precisely at which TPP or batch size that D2Z will first prevail; here we will rely on our empirical findings (Section 4) to fill in the theoretical gap.

**Hypothesis 2**: *As TPP increases, the optimal peak LR will decrease, for all LR schedules.*

Tuning peak LR is about trading off movement from initial conditions (requiring a high LR) and mitigating variance (requiring a low LR). As TPP increases, and bias reduction plays a smaller role, optimal peak LR should decrease. We hypothesize the decrease will be greater with a *Constant* schedule, as *Constant* does not use decay to balance the conflicting demands of bias and variance (Figure 3). Moreover, optimal peak LR for other *continuous* LR schedules, like *WSD* and *Cyclic*, should also decrease with longer training durations. In this way, such "schedule-free" approaches are *not truly schedule-free*. This dependence is also obvious when plotting update coefficients for these schedules (appendix Figure 29): the higher the LR, the more emphasis on recent updates.

**Hypothesis 3**: Linear *D2Z will improve over* Cosine *D2Z.*

While LR decay allows averaging over more weight updates, if the LR decreases too quickly, the final weight updates may not contribute to the EMA. From a loss surface perspective, as the LR approaches zero, we take vanishingly small steps. Since *Cosine* reaches smaller steps faster than *Linear* (Figure 2, left), *Cosine* will make less progress toward the optimum loss. From the EMA perspective, this is equivalent to *Cosine* having smaller $c_{t,i}$ coefficients as $i$ approaches $t$ (Figure 2, right). Note this problem is unique to *Cosine* D2Z and will not affect, e.g., *Cosine* 10× decay.

**Hypothesis 4**: *A high LR (not weight decay) reduces bias and achieves optimal loss.*

Note that weight updates $x_t$ have a coefficient of $1/\lambda$ in Equation (4). So, while $\eta_t$ and $\lambda$ contribute equally to $\alpha_j$, increasing $\lambda$ to reduce bias is counterproductive as weight updates will be scaled down proportionally, reducing movement from initial conditions. However, if LR is in a high enough range so that bias plays a minimal factor, both LR and WD should equally affect variance reduction.

## 4 EMPIRICAL ANALYSIS OF DECAY-TO-ZERO

### 4.1 EXPERIMENTAL SETUP

Experiments use a GPT-like LLM (Radford et al., 2019), with ALiBi embeddings (Press et al., 2022) and SwiGLU (Shazeer, 2020). Main paper models are trained on SlimPajama (Soboleva et al., 2023) and evaluated over 1.1B held-out tokens (regardless of training TPP). Unless otherwise indicated, $\lambda$=0.1 is used. Training runs use the same random seed, so all decay functions (*Linear*, *Cosine*, etc.) and ratios (*Constant*, 10×, D2Z) have identical warmups, but note results are very consistent across seeds at this scale (appendix Figure 24). By default we use $\mu$P (standard parameterization results are in Appendices C.2 and C.10). $\mu$P HPs are derived from a smaller proxy model tuned using a *Linear-*10× schedule. Since we hypothesized that different LR schedules may enjoy their optimal peak LR at different values, our experiments compare schedules when each is tuned to its optimal peak LR; we sweep $\tilde{\eta}$ by factors of 2× around the $\mu$P proxy-tuned $\tilde{\eta} = 1.6$e-02. Appendix A provides further details on the model architecture (Table 2), dataset sizes (Table 3), and compute resources.

### 4.2 RESULTS

**Finding 1**: Linear-*D2Z is the optimal LR schedule across virtually all peak learning rates.*

For 610M-parameter $\mu$P models trained to compute-optimal 20 TPP, the optimal *Linear*-D2Z setting achieves 0.77% lower loss than the optimal *Linear*-10× setting (Table 1). At smaller, suboptimal

Table 1: **Comparison of schedules**: validation loss for 610M $\mu$P models, 20 TPP. $\tilde{\eta}$=1.6e-02 is the $\mu$P proxy-tuned peak (base) LR, $\eta$ is the LR after $\mu$P adjustment ($\rho$=0.125, see Section 2.2). D2Z is superior to 10× decay across most LRs. *Linear*-D2Z slightly outperforms *Cosine*-D2Z in all cases.

| $\tilde{\eta}$ | $\eta$ | *InvSqrt* | *Constant* | *Cosine* 10× | *Linear* 10× | *Cosine* D2Z | *Linear* D2Z |
|---|---|---|---|---|---|---|---|
| 6.5e-02 | 8.1e-03 | 2.789 | 3.035 | NaN | 2.667 | 2.611 | **2.605** |
| 3.2e-02 | 4.0e-03 | 2.710 | 2.850 | 2.604 | 2.606 | 2.574 | **2.571** |
| 1.6e-02* | 2.0e-03 | 2.671 | 2.768 | 2.590 | 2.591 | 2.578 | **2.573** |
| 8.1e-03 | 1.0e-03 | 2.665 | 2.722 | 2.598 | 2.600 | 2.595 | **2.590** |
| 4.0e-03 | 5.1e-04 | 2.691 | 2.711 | 2.634 | 2.635 | 2.637 | **2.633** |
| 2.0e-03 | 2.5e-04 | 2.762 | 2.739 | **2.707** | 2.710 | 2.717 | 2.714 |

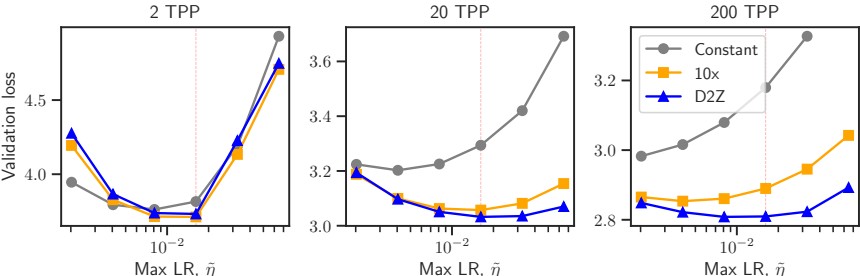

Figure 5: **Comparing decay schedules across TPP** (111M scale): As TPP increases, *Linear*-D2Z outperforms 10×, especially at the proxy-tuned peak LR (red lines).

peak LRs, 10× can be better than D2Z. In appendix Figure 10, we demonstrate very similar results using the standard parameterization. Regarding the decay function, gains from *Linear*-D2Z over *Cosine*-D2Z are small, but perfectly consistent across peak LRs, exactly in line with recent work (Defazio et al., 2023; Lingle, 2024). Interestingly, *Cosine*-10× is consistently slightly better than *Linear*-10×, showing that *Linear* itself is not always best, rather *Linear plus* D2Z is needed. Lacking a cooldown phase, inverse square root (*InvSqrt*) and *Constant* do not perform as well.

Appendix Figure 11 compares schedules as we sweep peak LR for 111M, 610M, and 1.7B models. For these and subsequent experiments, we use a *Linear* decay.

> **Finding 2**: *As TPP increases, the relative improvement of D2Z over 10× also increases.*

For 111M models at only 2 TPP, D2Z performs *worse* than 10× across all peak LRs (Figure 5, left). 610M results are similar (appendix Figure 12). However, as TPP increases, D2Z begins to outperform 10×, exceeding the best setting of 10× by 1.6% at 200 TPP, and performing 2.8% better at the proxy-tuned $\tilde{\eta}$=1.6e-02 (marked in plots with a red vertical line). While we hypothesized D2Z would surpass 10× at *some* TPP, our key finding is that, at both compute-efficient (20 TPP) and over-trained sizes, decaying to *zero* is optimal. Even at 2 TPP, some LR decay is beneficial (10× improves over *Constant*), but too much evidently hampers bias reduction. As TPP increases, optimal LRs for 10× shift lower as variance reduction gains in importance, but now it is the lower *peak* LRs that hamper bias reduction, and 10× begins to lag D2Z. By 20+ TPP, D2Z is consistently superior.

Figure 6 plots validation loss of 10× and D2Z at different TPP settings for different model sizes (all models are trained using the proxy-tuned peak LR). For the 610M model, D2Z is also initially worse than 10×, but begins to surpass it around 4 TPP, and by 200 TPP is 2.6% better. As with training loss (Figure 1), an 80 TPP D2Z model can surpass a 200 TPP 10× model in validation loss. **Importantly, these trends also hold for *downstream* evaluation of the models** (appendix Table 5).

Figure 6 also shows the advantage of using D2Z with *1.7B models trained to 80 TPP*; D2Z achieves roughly 3.05% lower loss than 10× at the proxy-tuned $\tilde{\eta}$. At this scale, such gains provide real-world impact: we over-train models of this size for use as proposal models in speculative decoding (Leviathan et al., 2023) and smaller models like Gemma-2B (Mesnard et al., 2024) and Phi-1.5 (Li et al., 2023) are used in many production settings.

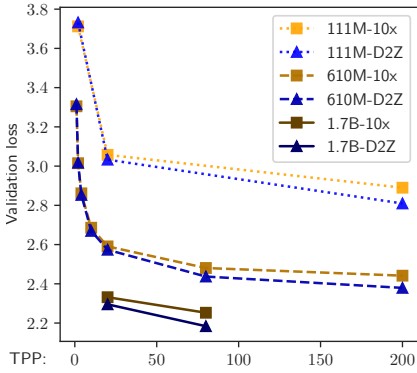

Figure 6: **Loss gap between *Linear-D2Z* and 10×** grows with TPP: for 111M, 610M, and 1.7B-scale models.

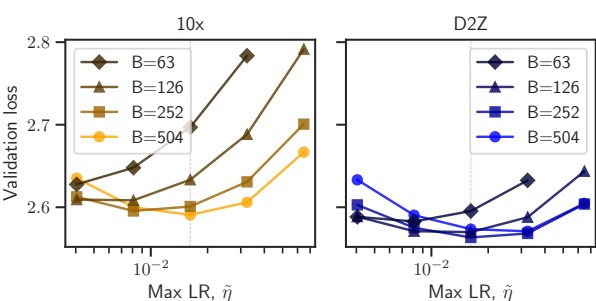

Figure 7: **Comparison across batch sizes (iso-TPP)** (610M): Comparing 10× (left) to D2Z (right). All models trained to 20 TPP (smaller batches have more steps). As $B$ decreases, the optimal LR ($\tilde{\eta}$) drifts significantly lower for 10× decay, less so for D2Z.

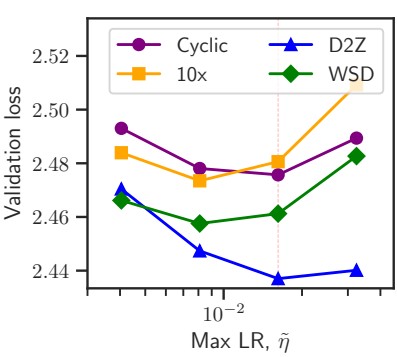

Figure 8: **Comparison to continuous** (610M, 80 TPP): D2Z surpasses *Cyclic* and *WSD* schedules.

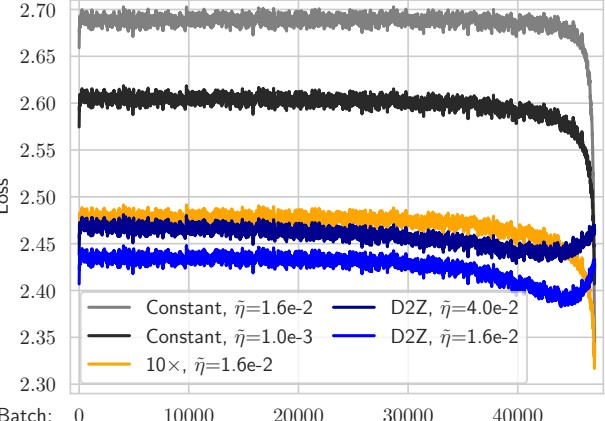

Figure 9: **Loss on *training set* batches** (in original order) after models have been trained (610M, 80 TPP)

> **Finding 3**: *Compared to* Constant *and 10×, optimal peak LRs are much more stable with D2Z.*

Figure 5 reveals different levels of hyperparameter sensitivity when increasing TPP: optimal LR shifts substantially lower for *Constant*, somewhat lower for 10×, and hardly at all for D2Z. The same trend holds at the 610M scale (appendix Figure 12). Moreover, with D2Z, loss is less sensitive to a suboptimal LR (bowls are flatter). D2Z loss is also more stable (and lower) as we vary both *weight decay* (appendix Figures 17 and 18) and *batch size*, $B$. For example, the optimal LR already shifts significantly for 10× models at B=126, while D2Z optima only begin to shift at B=63 (Figure 7). The superiority of D2Z over 10× across $B$ is more clearly observed in appendix Figure 13.

Crucially, if we keep the number of optimization steps constant (11752), but only vary $B$ (effectively increasing TPP proportional to $B$), we find the optimal learning rate does not *decrease* (i.e., with TPP) but rather *increases* (appendix Figure 14). Clearly, LR decay is primarily beneficial as a noise-reduction mechanism: with larger batches, we have less gradient noise, and can afford a larger LR.

> **Finding 4**: Linear-*D2Z works better than* WSD *and* Cyclic *(continuous) LR schedules.*

In Figure 8, we compare D2Z to 10×, and to two approaches designed for continuous pre-training: *Cyclic*, which cycles LR up and down, and *WSD* (Section 2.1). For *WSD*, we simulate a model being retrieved after 80 TPP, and cool the LR to zero for the final 22.5% of steps (around the proportion recommended by Hägele et al. (2024), and equal to the cooldown duration in our 20 TPP models). Appendix Figure 29 provides full LR curves and update coefficients for all models in Figure 8.

At its optimal LR, *WSD* works better than $10\times$, confirming results in Hägele et al. (2024). However, note the optimal peak LR shifts lower for both $10\times$ and *WSD* at this TPP. *Linear*-D2Z remains best here, around 0.84% better than the optimal *WSD*. Given the diminishing returns of high-TPP training (Figure 6), *WSD* would require significantly more training FLOPs to reach the level of D2Z.

> **Finding 5**: Constant *and* $10\times$*, but not D2Z, strongly overfit to the end of the training data.*

Figure 9 shows the loss of trained models (with frozen weights) on the exactly-same ordered batches used during pre-training. As predicted by the extended EMA perspective (Section 3), the higher the LR, the more models fit to late-stage training batches. It is striking that both *Constant* and $10\times$, but not D2Z, *overfit* to the very final portion. Extra adaptation of generative models to recent training sequences has long been observed (Graves, 2013), but to our knowledge this is the first evidence D2Z may help mitigate these effects. Since D2Z demonstrates the lowest loss on batches *slightly before* the final training phase, placing the highest-quality and most-recent data in the *very* final phase, while using D2Z (e.g., as in Dubey et al. (2024)), may be suboptimal. These findings also contradict Biderman et al. (2023), who found training order had little impact on memorization.

## 5  DISCUSSION

**The confounding role of LR schedule**   We have shown that the more constant the LR schedule, the more sensitive the optimal peak $\eta$. These findings help explain LR sensitivity in prior work. For example, Shen et al. (2024b) were puzzled by their observation that "although the *WSD* scheduler could, in theory, continue in the stable phase forever ... the optimal LRs are different for different amounts of training tokens." As noted under Hypothesis 2, LR schedules with long constant periods only appear "schedule-free" in the primal; from the dual (EMA) perspective, the higher the LR, the more emphasis is placed on recent updates (see appendix Figure 28a for *Constant*, Figure 29d for *WSD*). Filatov et al. (2024) and Yang et al. (2021, Figure 19) also observed major decreases in optimal LR with higher TPP; these results were both obtained with a constant LR. In contrast, Bjorck et al. (2024) observed less significant decrease of optimal $\eta$ with TPP, but used $10\times$ decay.

LR schedule may also affect the optimal $\eta$ as batch size varies — particularly when using $\mu$P. Some prior $\mu$P studies (Yang et al., 2021; Noci et al., 2024; Shen et al., 2024b) observed linear scaling of optimal LR with batch size, i.e., the so-called *linear scaling rule* (Krizhevsky, 2014; Chen et al., 2016; Smith et al., 2018). Others (Lingle, 2024) have observed square-root scaling, resonating with other prior studies (Hoffer et al., 2017; You et al., 2019b; Malladi et al., 2022). This discrepancy can be explained by noting the linear scaling results were all found with a *Constant* or *WSD* LR decay, while square-root was observed with *Linear* D2Z, again underscoring the greater stability of D2Z.

**The confounding role of training duration**   While LR schedules have confounded studies varying TPP, TPP has analogously confounded studies evaluating LR schedules. Recall Kaplan et al. (2020) also saw a benefit from D2Z. In contrast to Chinchilla scaling (Hoffmann et al., 2022), in the Kaplan et al. perspective, small models should be trained to high TPP (while larger models should be trained less). It is therefore not surprising Kaplan et al. saw benefits testing D2Z with small models; as we have shown, D2Z is especially effective at high TPP. In contrast, in Figure 4 of Yang et al. (2021), *Linear* is *worst* of all schedules, and the gap between it and *Constant* and *InvSqrt* grows with model width. But here, since training data is small — and *fixed*, then TPP *decreases* as width increases. Thus training is in a phase where bias reduction is paramount and D2Z is not effective. Notably, in their LLM training experiments at higher TPP, Yang et al. do report linear D2Z to work best.

With this context, we speculate on why D2Z has not been adopted more widely in LLM pre-training:

- First, it is common to evaluate hyperparameters on smaller training runs; unfortunately, with limited training data (low TPP), D2Z misleadingly under-performs.

- Secondly, coupling between LR schedule and optimal peak LR is problematic: with $10\times$ decay, we may find a lower peak LR is optimal; if we then test D2Z with the same LR, it may be suboptimal for D2Z. Not seeing a benefit at this LR, we may conclude D2Z is inferior in general.

- Finally, poorly controlled training dynamics may prevent D2Z models from training with their (higher) optimal LR. Indeed, when we initially compared D2Z and $10\times$ using NanoGPT (Appendix C.10), $10\times$ performed better at the default LR. Raising the LR resulted in training diver-

gence. Only after switching the numerical precision from `bfloat16` to `float32` could D2Z succeed — and the model reach optimal loss.

**The special benefit of D2Z** Our results suggest a low $\alpha$ later in training can expand the timescale over which weight updates are combined, reducing noise in a similar way to increasing batch size. However, there is apparently a separate, independent benefit from a vanishing LR. Indeed, looking at appendix Figure 29 for the 80 TPP comparison to continuous schedules, we see $10\times$ coefficients are quite similar to the D2Z curve, apart from missing the final drop. Moreover, they are flatter than the *WSD* EMA coefficients for the same peak LR, suggesting better integration of prior updates. Yet *WSD* performs *better* than $10\times$ at all LR settings (Figure 8). In contrast, at 2 TPP (Figure 5, left), $10\times$ performs better than D2Z at every LR setting. Prior work has shown large LRs allow exploration of the loss surface at a height "above the valley floor" (Xing et al., 2018), while LR cooldown phases descend into a local minimum (Hu et al., 2024; Hägele et al., 2024). It appears that descending into these minima is beneficial only after sufficient exploration of the loss surface.

## 6 LIMITATIONS AND FURTHER EXPERIMENTS

While our findings strongly support linear D2Z being optimal in our specific context, there are some limitations to keep in mind. First, LR schedules like D2Z require prior knowledge of the total number of training steps. But it is worth reiterating that even nominally schedule-free approaches such as *Constant* also require this knowledge in order to optimally set the LR value. In contrast, the extended EMA perspective of LR schedules enables derivation of a truly schedule-free schedule, which we introduce in Appendix D.1. Second, our focus in this paper was specifically LLM training at compute-optimal and overtrained dataset sizes. For ML problems with limited access to training data, D2Z is likely not the best strategy. Third, our work focuses on AdamW (the standard optimizer in LLM pre-training). While the extended EMA perspective of LR schedules will likely apply to other optimizers that use decoupled weight decay (as noted by Wang & Aitchison (2024)), it may not apply to approximate second order methods, such as Shampoo (Gupta et al., 2018). Finally, for LLMs with unstable training dynamics that cannot tolerate high LRs, D2Z may not be beneficial. We experienced this first-hand when we initially trained NanoGPT (Appendix C.10).

However, we also note the remarkable consistency of D2Z's success. While results in the main paper used the SlimPajama dataset, consistent results were found with OpenWebText (Appendix C.10) and a multilingual dataset (with a larger vocabulary) (Appendix C.11). We also saw consistent findings with different parameterizations (Appendices C.2 and C.10), architectures (Appendix C.11), weight sparsity settings (Appendix C.8), and training frameworks (Appendix C.10). Appendix C.11 describes scaling laws fit to models trained with D2Z and $10\times$, at model sizes up to 2.75B; results indicate a growing performance gap as scale increases. Finally, Appendix C.6 demonstrates that the benefits of *weight decay* are observed primarily when using LR D2Z, where raising weight decay can fine-tune the EMA update coefficients without affecting training stability.

## 7 CONCLUSION

Linear decay-to-zero is the optimal LR schedule for LLM training using AdamW. To be clear, less decay is beneficial for low tokens-per-parameter training, but there is no practical reason to perform such training with LLMs, since the same FLOPs could be used to train a smaller model, over more tokens, to a lower loss – using D2Z. The superiority of D2Z was validated across a range of experimental conditions. Results suggest its relative benefit will increase as models increase in scale. Moreover, when using D2Z and $\mu$P, the optimal peak LR is less sensitive to changes in weight decay, dataset size, and batch size, i.e., there is better hyperparameter *transfer*.

Our analysis was aided by our interpretation of AdamW's output as a convex combination of prior weight updates. D2Z overfits less the final training sequences, and is especially beneficial when gradient noise dominates training. As we enter a phase of applied ML where inference efficiency is a primary concern, there is strong motivation to study high-TPP training, where gradient noise is the bottleneck. While our results indicate that D2Z is a key component of the solution here, further investigation is required, including into how and when to adjust hyperparameters such as weight decay, batch size, and learning rate, in the high-TPP context.

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

Table 2: **Model architecture and batch sizes for main experiments**

| Model | vocab. size | $d_{model}$ | $n_{layers}$ | $d_{head}$ | $d_{\text{ffn}}$ | batch size |
|-------|-------------|-------------|--------------|------------|------------------|------------|
| 111M  | 50257       | 768         | 10           | 64         | 2048             | 192        |
| 610M  | 50257       | 2048        | 10           | 64         | 5461             | 504        |
| 1.7B  | 50257       | 2048        | 32           | 64         | 5461             | 504        |

Table 3: **Training steps for main experiments**

| Model | TPP | Warmup | Steps  | Tokens |
|-------|-----|--------|--------|--------|
| 111M  | 2   | 56     | 557    | 219M   |
| 111M  | 20  | 556    | 5568   | 2.19B  |
| 111M  | 200 | 5560   | 55680  | 21.9B  |
| 610M  | 2   | 118    | 1176   | 1.21B  |
| 610M  | 20  | 1175   | 11752  | 12.1B  |
| 610M  | 80  | 4700   | 47008  | 48.5B  |
| 610M  | 200 | 11750  | 117520 | 121B   |
| 1.7B  | 20  | 3322   | 33220  | 34.3B  |
| 1.7B  | 80  | 13288  | 132880 | 137B   |

# A    EXPERIMENTAL DETAILS

Table 2 provides details on model architecture and hyperparameters for the main experiments (i.e., results presented in the main paper). Table 3 provides information on the training steps. All the models in our main experiments were trained on the SlimPajama dataset (Soboleva et al., 2023), a cleaned and deduplicated version of the RedPajama dataset. We use the GPT-2 (Radford et al., 2019) vocabulary of size 50257, and a context length of 2048 tokens. Unless otherwise noted, the weight decay value, $\lambda$, is by default set to 0.1, following standard practice in LLM pre-training (Brown et al., 2020; Hoffmann et al., 2022; Almazrouei et al., 2023; Aleph Alpha, 2024). Also following standard practice, we do not apply weight decay or bias to LayerNorm layers. Validation loss is always computed over 1.1B held-out tokens, regardless of training TPP. By default we parameterize with $\mu$P (further details below).

For a given TPP, all models have the exact same warmup phase: a linear warmup of the LR from 0 to the peak value. In all our runs, warmup was 10% of the total steps. LR warmup is standard practice in LLM pre-training.[3]

All models in the main experiments were trained on a Cerebras CS-3 system. 610M-parameter, 20 TPP models take roughly 6 hours each to train on a single CS-3. If a training run did not complete due to numerical instabilities, the values are left off our plots or marked as `NaN` in Table 1.

**Proxy model hyperparameter tuning**    To find the optimal $\mu$P hyperparameters (HPs), we trained a 39M-parameter proxy model using a width $d_{\text{model}}=d_p$ of 256, with 24 layers and a head size of

---

[3]While prior work has suggested LR warmup is less valuable in modern Pre-LN Transformers (Xiong et al., 2020), various other studies have shown warmup leads to lower loss (Goyal et al., 2018; Liu et al., 2019; Tissue et al., 2024; Kosson et al., 2024), and may reduce sensitivity to peak LR (Wortsman et al., 2023). In light of the similar benefits of D2Z, it would be interesting to investigate the value of warmup for models that are specifically trained using D2Z.

Table 4: **Tuned hyperparameters for $\mu$P proxy model**

| | |
|---|---|
| $\sigma_{W,\text{base}}$ | 8.67e-02 |
| $\tilde{\eta}$ | 1.62e-02 |
| $\alpha_{\text{input}}$ | 9.17 |
| $\alpha_{\text{output}}$ | 1.095 |

Table 5: **Downstream evaluations**: for 610M-parameter models corresponding to Figure 1. A model trained for 80 tokens-per-parameter with linear D2Z has equivalent downstream loss to the same model trained for 200 TPP with 10× decay.

| | MMLU (Avg.) | | Commonsense Reasoning | | | | | | | | Reading Comp. | Truthfulness & Bias | | Down-stream (Avg.) |
|---|---|---|---|---|---|---|---|---|---|---|---|---|---|---|
| | Wino-grande | Hella-swag | Open-Book QA | Lamb-ada OpenAI | Lamb-ada Stand. | SIQA | PIQA | Arc-e | RACE | | Truth-ful QA | CrowS-Pairs | |
| 10×@80TPP | 23.6% | 52.2% | 43.9% | 31.4% | 46.7% | 36.7% | 32.8% | 68.7% | 47.6% | 32.3% | 39.8% | 60.7% | 43.05% |
| D2Z@80TPP | 23.5% | 53.4% | 44.6% | 31.6% | 46.9% | 37.3% | 33.2% | 68.8% | 48.8% | 33.4% | 40.2% | 60.8% | 43.54% |
| 10×@200TPP | 23.3% | 53.4% | 46.6% | 31.2% | 46.2% | 38.8% | 32.2% | 68.8% | 47.9% | 34.4% | 38.4% | 60.4% | 43.46% |
| D2Z@200TPP | 24.7% | 54.5% | 48.2% | 32.4% | 50.0% | 42.6% | 32.9% | 70.1% | 50.4% | 32.6% | 38.9% | 62.5% | 45.00% |

64. We trained this proxy model on 800M tokens with a batch size of 256 and context length 2048, using 10× decay. We randomly sampled 350 configurations of base learning rates, base initialization standard deviation, and embedding and output logits scaling factors, and used the top-performing values as our tuned HPs (Table 4).

## B DERIVATIONS

### B.1 DERIVATION OF EQUATION (6)

The initial coefficient is $c_{t,1} = \prod_{j=2}^{t}(1 - \alpha_j)$. Clearly, if $\alpha_j = \alpha$ is constant, we have $c_{t,1} = (1-\alpha)^{t-1}$. Otherwise, given $\alpha_j$ is small, we can use a first-order Taylor expansion, $e^{-\alpha_j} \approx 1 - \alpha_j$, and therefore:

$$c_{t,1} \approx \prod_{j=2}^{t} e^{-\alpha_j}$$
$$= e^{\sum_{j=2}^{t} -\alpha_j}$$
$$= e^{-\sum_{j=2}^{t} \alpha_j}.$$

Assuming we only know the average value, $\bar{\alpha} = \frac{1}{t-1} \sum_{j=2}^{t} \alpha_j$, we have:

$$c_{t,1} \approx e^{-\bar{\alpha}(t-1)}$$
$$= (e^{-\bar{\alpha}})^{t-1}.$$

Given $\bar{\alpha}$ is also small, we reverse the earlier Taylor expansion, but now $e^{-\bar{\alpha}} \approx 1 - \bar{\alpha}$, and:

$$c_{t,1} \approx (1 - \bar{\alpha})^{t-1}.$$

That is, with a small time-varying smoothing parameter, the initial coefficient in an EMA, $c_{t,1}$, also decreases exponentially with the number of steps.

## C ADDITIONAL EXPERIMENTAL RESULTS

In this section, we include some additional results to support the findings in the main paper. All validation losses reported in this section are from models trained with *Linear* decay.

### C.1 DOWNSTREAM EVALUATIONS

**Finding 6**: *Trends in validation loss also hold for downstream evaluation.*

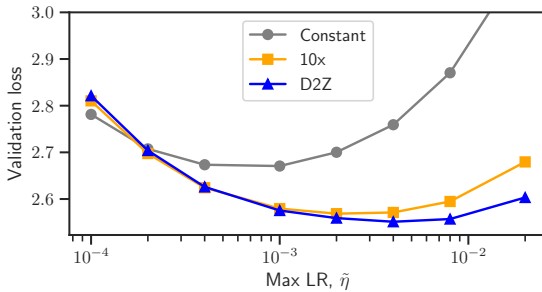

Figure 10: **Standard parameterization results**: Validation loss for different LR and decay combinations, across peak LRs, for 610M-parameter models trained with the standard parameterization.

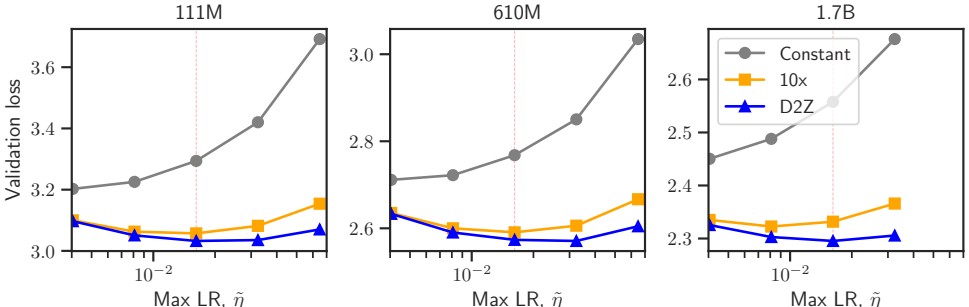

Figure 11: **Model size comparison**: Validation losses at 20 TPP. Across all model sizes, *Linear*-D2Z outperforms *Linear*-10×. Note: Missing high-LR values in all plots correspond to failed training runs due to `NaN` instabilities.

Table 5 presents a variety of downstream evaluations of the four models presented in Figure 1. Differences between the models here are largely consistent with the differences in training and validation loss, showing that D2Z is meaningful not just for the autoregressive training objective, but for real-world applications.

## C.2 STANDARD PARAMETERIZATION

**Finding 7**: *Similar trends hold for the standard parameterization.*

Figure 10 presents results for a 610M-parameter model trained with the standard parameterization. Here $\tilde{\eta}$ is therefore not a $\mu$P-corrected base LR, but rather a LR that we swept directly for this model scale. Results are obviously quite similar to results using $\mu$P, suggesting the benefits of D2Z are not $\mu$P-specific. Further results using the standard parameterization, but for NanoGPT models, are in Appendix C.10 below.

## C.3 MODEL SIZES

**Finding 8**: *Improvement of D2Z over 10× is greater at 1.7B scale.*

Figure 11 presents results across 111M, 610M, and 1.7B model sizes, all trained to 20 TPP. Note the absence of results for the highest LR setting at the 1.7B-scale; at the very highest LR, numerical instabilities led to failed training runs. Otherwise, results are fairly similar across model sizes. At the proxy-tuned peak LR, the gap between D2Z and 10× is 0.81%, 0.67%, and 1.56%, at the 111M, 610M, and 1.7B scales, respectively. We further investigate the issue of whether the gap between D2Z and 10× varies with model size as part of our scaling law experiments below (Appendix C.11).

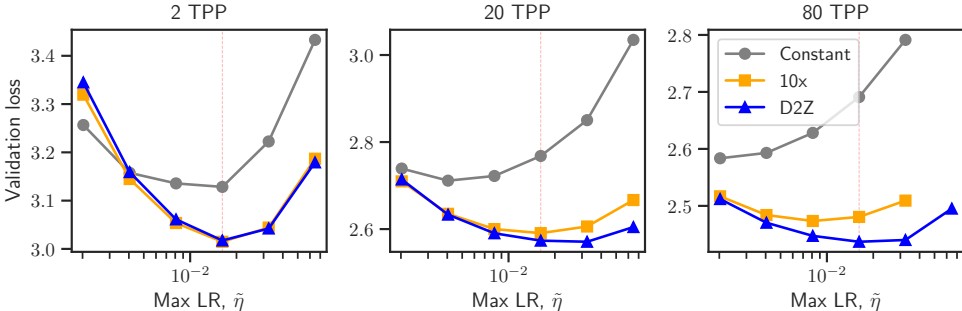

Figure 12: **Comparing decay schedules across TPP** (610M scale): As TPP increases, *Linear*-D2Z begins to outperform *Linear*-10×, especially at the proxy-tuned peak LR (red lines). The optimal LR also shifts significantly lower for *Constant*, somewhat lower for 10×, and hardly at all for D2Z. Compare to Figure 5 for 111M models.

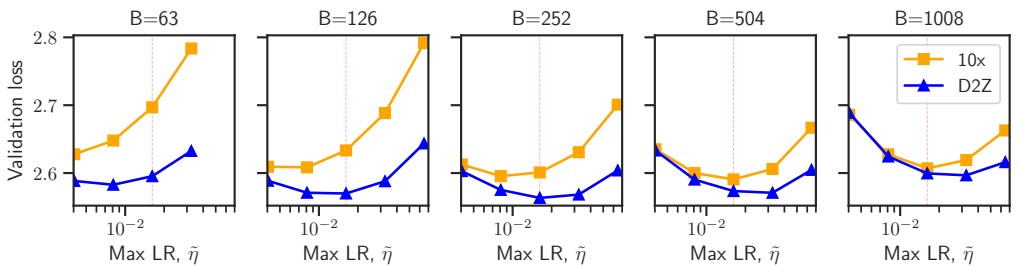

Figure 13: **Comparison across batch sizes (iso-TPP)** (610M): Different *batch sizes*, $B$, but *all trained to 20 TPP*. As batch size decreases, the relative gain of D2Z over 10× increases. Same data as in Figure 7.

## C.4 TPP

> **Finding 9**: *Improvement of D2Z over 10× grows with TPP (610M scale).*

As we vary TPP, we consistently see increasing gains with D2Z. Here we plot the results for the 610M-scale models in Figure 12, as a counterpart to main Figure 5.

## C.5 BATCH SIZES

**Batch size setup** Default batch sizes of 192 and 504 (Table 2) were selected for 111M and 610M scales. These specific values were based on an internal scaling law for optimal batch size as a function of compute FLOPs (similar to those in Bi et al. (2024) and Porian et al. (2024)). Batch size of 504 was then re-used for 1.7B training (later testing confirmed this to be a good setting at 1.7B for 20 TPP training). In our experiments varying batch size, we swept $B$ by factors of 2 around these initial settings.

### C.5.1 ISO-TPP BATCH SIZE EXPERIMENTS

> **Finding 10**: *Improvement of D2Z over 10× grows as batch size decreases (at constant TPP).*

In main paper Figure 7, we presented results where all models train for the same number of total *tokens* (20 TPP), while only the batch size, $B$, varies. Figure 13 presents the same data, but with each $B$ separated into a separate subplot; this lets us better observe how the differences between D2Z and 10× evolve as $B$ changes. We see clearly that for smaller $B$, as gradient noise increases, the differences between D2Z and 10× also increase. This again demonstrates that LR decay is beneficial as a noise reduction mechanism.

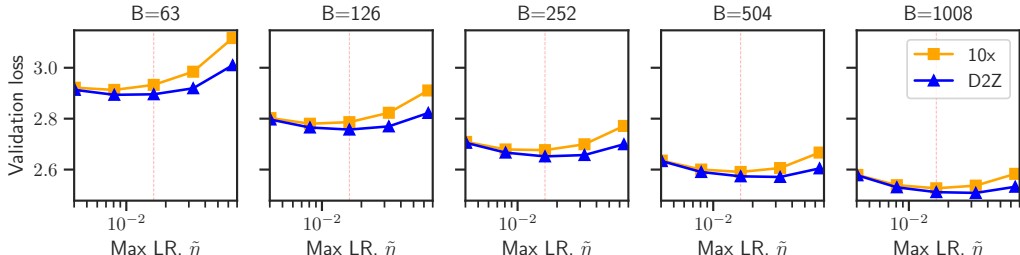

Figure 14: **Comparison across batch sizes (iso-step)** (610M): Different batch sizes, $B$, but all trained for *11752 steps* (not iso-FLOP, smaller batches see fewer TPP). In contrast, Figure 7 and appendix Figure 13 instead keep *TPP* constant. D2Z remains superior in the iso-Step context. More interestingly, here the optimal LR actually *increases* as batches — and, correspondingly, the *datasets* — increase in size.

We now discuss the related observation that the optimal LR decreases as $B$ decreases. Note the EMA perspective of Wang & Aitchison (2024) would predict linear scaling of $\tilde{\eta}$ with $B$. This is because the total number of iterations scales with $1/B$. Therefore, in order to keep the timescale constant (as a fraction of the total iterations), we would need to scale $\tilde{\eta}$ (or $\lambda$) proportional to any change in $B$. Although we do see roughly linear scaling in the case of $10\times$ models, the optimal $\tilde{\eta}$ is more stable with D2Z. In our conceptual model, the purpose of the timescale is to optimize variance reduction. But note that lowering $\tilde{\eta}$ also has an effect on bias reduction. For $10\times$, as $B$ changes, it seems paying the cost of lower bias reduction is worth the benefits in variance reduction. Since using D2Z already enables better variance reduction, it seems, for D2Z, lowering $\tilde{\eta}$ has less value in the trade-off, so optimal $\tilde{\eta}$ lowers to a lesser extent.

### C.5.2 ISO-STEP BATCH SIZE EXPERIMENTS

> ***Finding 11****: Optimal LR increases with batch size when training over the same number of batches/steps.*

In the above batch size tests, we kept the dataset size constant (iso-TPP). Now consider the following two mechanisms for increasing the dataset size (TPP) of a training run:

1. Fix the batch size, but increase the number of steps.
2. Fix the number of steps, but increase the batch size.

Approach (1) was taken in our experiments increasing TPP. In that case, we found optimal LR to decrease, as a mechanism for coping with greater gradient noise, which grows with TPP. We now discuss Approach (2): fixing the number of steps, but increasing the batch size (iso-step).

In Figure 14, we keep the number of *steps* constant to 11752, so each model will see the same total number of batches, and we increase $B$ by factors of two.[4] Since number of steps does not change, Wang & Aitchison (2024) do not prescribe any adjustments to the timescale, and hence no adjustment is needed to LR (or weight decay). But we clearly see that optimal LR increases as $B$ increases (like a rolling wheel, LR bowls rotate clockwise as we move to the right). This is evidently because increasing $B$ decreases gradient noise; with less noise, we can afford a larger LR throughout training. More precisely, increasing $\tilde{\eta}$ decreases the number of AdamW weight updates that are combined. Therefore, an increase in $\tilde{\eta}$ can be viewed as a way to reduce the effective number of batches that are combined, basically compensating for the fact per-step $B$ is larger.

As $B$ increases, gradient noise decreases, and we would expect benefits of LR decay to also diminish; differences between $10\times$ and D2Z do seem to decrease. This aligns with prior work, e.g., You et al. (2019a) compared SGD to full-batch GD (with WideResNets for image classification), and showed that when gradient noise is eliminated by using full GD, LR decay is no longer beneficial.

---

[4]Note, for purposes of scale, results at $B$=504 are the same in Figure 13 and Figure 14; the latter just has a larger range on the y-axis.

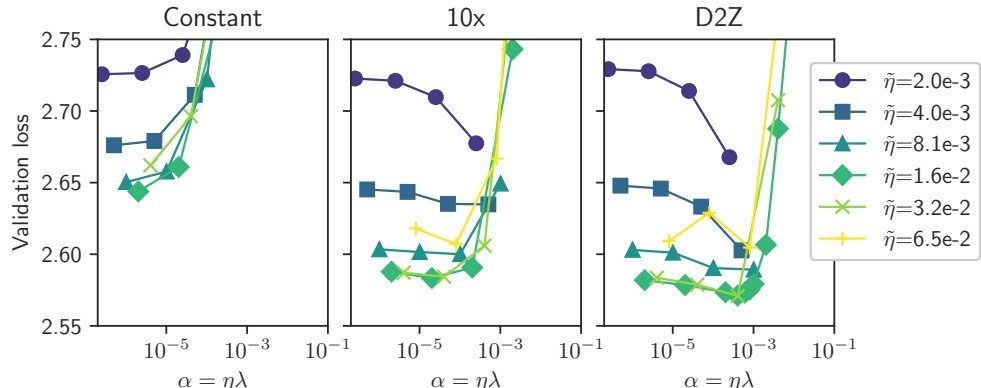

Figure 15: **Weight decay results, grouped by $\tilde{\eta}$ as $\alpha=\eta\lambda$ varies** (610M, 20 TPP): Validation losses for different settings of decay schedule (separated by subplot), maximum learning rate ($\tilde{\eta}$, marked by color), and weight decay ($\lambda$, corresponding to points on the LR curves). Loss is plotted relative to $\alpha=\eta\lambda$. The optimal loss corresponds to high (but not too high) $\tilde{\eta}$. As the smoothing $\alpha$ increases, *Constant* models suffer, $10\times$ models see marginal gains, while D2Z models benefit significantly.

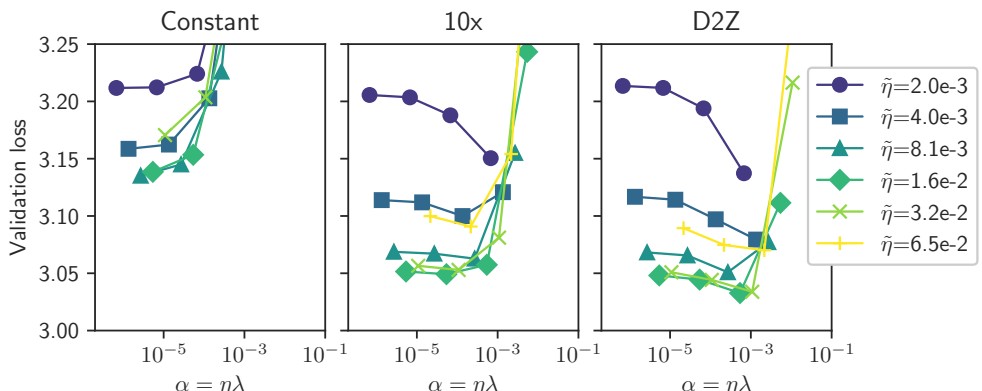

Figure 16: **Weight decay results, grouped by $\tilde{\eta}$ as $\alpha=\eta\lambda$ varies** (111M, 20 TPP): Validation losses for different settings of decay schedule (separated by subplot), maximum learning rate ($\tilde{\eta}$, marked by color), and weight decay ($\lambda$, corresponding to points on the LR curves). Loss is plotted relative to $\alpha=\eta\lambda$. The optimal loss is obtained when $\tilde{\eta}$ is high, but not too high, typically around 1.6e-02 to 3.2e-02. As in Figure 15, only D2Z models further improve as $\lambda$ increases.

## C.6 WEIGHT DECAY

### C.6.1 WEIGHT DECAY: RESULTS

***Finding 12****: Benefits of weight decay are observed primarily when using LR D2Z.*

Figure 15 analyzes the interaction between *weight decay* (WD) and LR for 610M models. Here we plot with the x-axis set to the (peak) smoothing parameter, $\alpha=\eta\lambda$ (on a log scale), and y-axis set to validation loss. Note that for a given decay ratio and fixed $\alpha$, EMA update coefficients (Section 3) are identical (i.e., specific values of $\eta$ and $\lambda$ do not affect coefficients, only their product). Thus it supports the extended EMA perspective to note that regardless of peak LR, $\tilde{\eta}$, models tend to reach lowest loss at around the same $\alpha$ (Figure 15); that is, the same EMA timescale, $1/\eta\lambda$, is optimal across different LRs. However, to reach optimal loss, models require $\tilde{\eta}$ to be high enough, but not too high — at $\tilde{\eta}$=6.5e-02 and above, we consistently encounter instabilities in training.

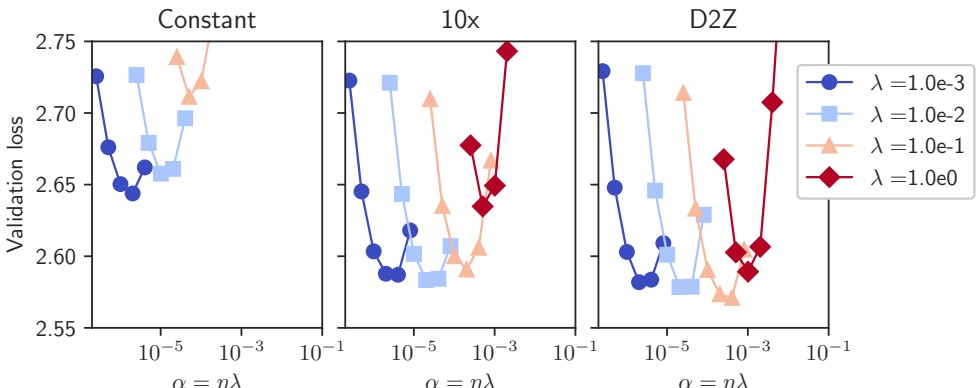

Figure 17: **Weight decay results, grouped by $\lambda$ as $\alpha=\eta\lambda$ varies** (610M, 20 TPP): Subset of the data in Figure 15, but now curves trace points with same weight decay $\lambda$ (in color) and LR $\eta$ varies across each curve. Only D2Z models significantly improve as we increase weight decay. D2Z models are also less sensitive to choice of $\lambda$.

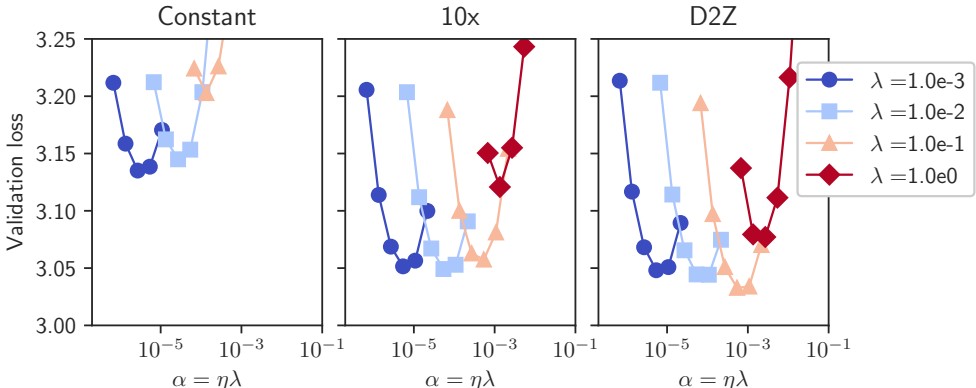

Figure 18: **Weight decay results, grouped by $\lambda$ as $\alpha=\eta\lambda$ varies** (111M, 20 TPP): Same data as in Figure 16, but now curves trace points with same weight decay $\lambda$ (in color) and LR $\eta$ varies across each curve. Only D2Z models significantly improve as we increase weight decay. D2Z models are also less sensitive to choice of $\lambda$. See Figure 17 for 610M-model results.

When increasing $\lambda$ with a given peak LR (moving left-to-right along curves), note behavior differs depending on LR schedule: *Constant* models perform worse, *Linear*-10× sees marginal gains, and *Linear*-D2Z benefits significantly. Results are similar for 111M models (Figure 16).

Figures 17 and 18 depict the same data, but with points now grouped by $\lambda$.[5] For higher or lower $\lambda$, we cannot reach the optimal timescale without taking $\tilde{\eta}$ out of its comfort zone. Therefore, with our 20 TPP models, $\lambda$=0.1 emerges as good default setting because, in synergy with a comfortable LR, $\lambda$=0.1 corresponds to an appropriate EMA timescale.

Similar to the findings in Andriushchenko et al. (2023), we also confirm weight decay does not act like a traditional regularizer here: beneficial settings of weight decay always improve both validation *and* training loss.

### C.6.2 WEIGHT DECAY: FURTHER DISCUSSION

**Learning rates reduce bias, weight decay best controls variance**    Recall that Hypothesis 4 made the observation that while LR $\eta$ and weight decay $\lambda$ can both equally change the update coefficients,

---

[5]Note Figure 17 excludes some points from Figure 15: Figure 15 includes some additional $\lambda$ settings specifically for $\tilde{\eta}$=1.6e-02, and we leave those off this figure in order to reduce clutter.

only $\eta$ is effective in moving the model away from initial conditions (due to the $1/\lambda$ factor applied to the updates). This is confirmed by Figures 15 and 16: only when $\eta$ is sufficiently high do models achieve optimal loss. Given such a high-enough $\eta$, the primary benefit of weight decay is evidently to adjust $\alpha=\eta\lambda$ (and hence the timescale $1/\alpha$), to a setting that synergizes best with the decay schedule (for 610M models, around 4e-04). Weight decay is specifically useful in this context because training instabilities prevent reaching this optimal $\alpha$ purely through increasing $\eta$. However, given $\eta$ and $\lambda$ both affect the timescale, $\alpha=\eta\lambda$ can be viewed as the *effective* or *intrinsic* LR (Li et al., 2020; Wang & Aitchison, 2024), but only later in training.

Since weight decay does not impact bias reduction, it follows that, when the timescale needs to be adjusted at *high TPP* due to the greater gradient noise, it should be more effective to lower WD than to lower LR, exactly as prescribed by Wang & Aitchison (2024). In ongoing concurrent work, we do observe lower loss from tuning WD compared to tuning LR. This stands in contrast to other recent work that recommends purely decreases in LR for high-TPP training (Shen et al., 2024b; Bjorck et al., 2024).

**Why weight decay is only beneficial when using LR decay**    Our observation that weight decay is only beneficial when using a LR (decay) schedule aligns with both Loshchilov & Hutter (2017) and Andriushchenko et al. (2023) who also observed no benefit from WD when using constant LRs. In contrast, Aleph Alpha (2024) recently reported $\lambda=0.1$ to improve over $\lambda=0.0$ in LLM training; notably, they train with D2Z. We believe the reason for these observations can also be explained by the bias/variance trade-off, and by the fact, noted above, that weight decay primarily plays a role later in training. Since its LR is fixed, *Constant* negotiates the bias/variance trade-off by using a LR that is suboptimally *low* early and suboptimally *high* later on (refer back to Figure 3 in the main paper for a visualization). Increasing $\lambda$ raises the already-too-high (late) $\alpha$ even higher, further decreasing the timescale $1/\alpha$ and hurting the model. In contrast, when using a LR schedule, weight decay can play its beneficial variance reduction role, as discussed above.

It is also worth observing that if one is using D2Z with a LR that is suboptimally low (e.g., if increasing it any further caused numerical stability issues, as we observed with NanoGPT in Appendix C.10), then increasing weight decay can be very beneficial. For example, note the large improvements in loss as weight decay is increased for the $\tilde{\eta}=4.0$e-3 and $\tilde{\eta}=2.0$e-3 curves in Figure 15.

**Stability of D2Z loss with high $\lambda$**    While *Constant* and $10\times$ are much worse at higher $\lambda$ values, D2Z performs reasonably well even at $\lambda=1.0$ (particularly at the 610M scale). Since increasing $\lambda$ effectively decreases the timescale $1/\alpha$ over which weight updates are integrated, it increases the variance over weight updates. Therefore, setting $\lambda$ too high hurts all models, but since D2Z has lower $\eta$ later in training, it somewhat counterbalances the increase in $\lambda$ in the product $\alpha=\eta\lambda$. Another view of this is that as fewer updates are combined, noise increases; D2Z is evidently best at mitigating the negative effects of such noise through its maximal LR decay.

## C.7    STEP DECAY

> **Finding 13**: *Step decay helps* Constant*, but does not provide benefits if already decaying the LR.*

In Figure 19, we present results investigating the impact of *Step* decay on model training. Here, for 111M-parameter models, *Step* decay can improve the loss versus keeping the LR constant (left panel), but the resulting losses are still much worse than those obtained with D2Z or $10\times$ decay. We also tried applying a *Step* decay to a LR that had been following a $10\times$ (middle panel) or D2Z trajectory (right panel); this approach always led to inferior results.

While it is likely possible to improve the quality of *Step* decay by tuning the positioning of the drop, we hypothesize that these efforts will not surpass D2Z, since dropping the LR will fundamentally always result in an over-emphasis of earlier updates, as shown in Figure 2 and Figure 27. Moreover, introducing additional tunable hyperparameters (i.e., when and how much to decay) is a further drawback of the *Step* schedule.

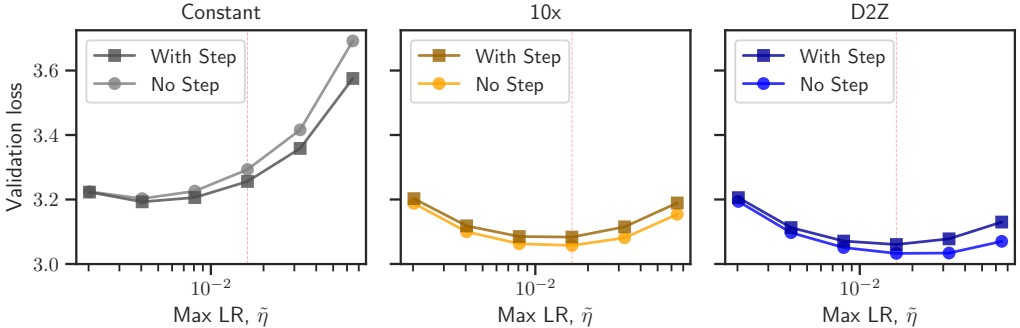

Figure 19: **Impact of applying *Step* decay** (111M, 20 TPP): Validation loss for different LR and decay combinations *with and without* Step *decay*. *Step* decay is applied after 90% of training, stepping to a value equal to 0.1% of the maximum LR. Dropping the LR in this manner helps *Constant*, although it is still below the level of 10× and D2Z. Adding stepping to a 10× or D2Z schedule is not beneficial.

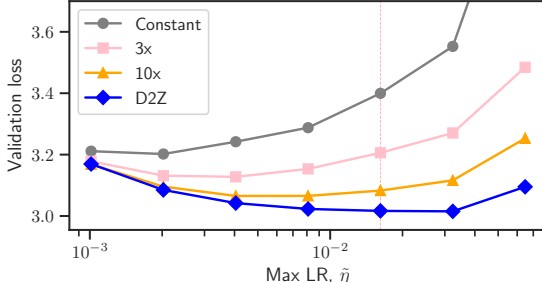

Figure 20: **Comparison of decay schedules when using 93.75% weight sparsity** (610M): Validation losses for different LR and decay combinations, for a model trained with SμPar, with 93.75% unstructured weight sparsity. As the decay rate increases, loss improves, while the optimal peak LR is much more stable when using D2Z.

## C.8    WEIGHT SPARSITY

> ***Finding 14**: Relative improvement of D2Z over 10× increases *with weight sparsity*.*

**Weight sparsity setup**    In this section, we investigate the role of *Linear*-D2Z in the context of models trained with *unstructured weight sparsity*, a promising direction for improving the efficiency of large neural networks (Hoefler et al., 2021). We parameterize with μP's sparse extension, SμPar (Dey et al., 2024b). SμPar allows us to use the same μP hyperparameters as with dense models, except we must now apply corrections due to both model scaling (i.e., $\rho$, Section 2.2) and layer sparsity. For these experiments, we sparsified all non-embedding layers of our 610M-parameter dense models by randomly fixing certain weights to zero for the duration of training. We trained all models for 11752 total steps using a batch size of 504, i.e., the same amount of training data as we used for training the corresponding 610M-parameter dense models to 20 TPP (Tables 2 and 3).

At 93.75% sparsity ($^1/_{16}$ density), the optimal D2Z model improves by 1.64% over the optimal 10× model, with a clear trend of optimal LRs shifting lower and loss becoming worse as we go from 10× to 3× to *Constant* decay (Figure 20).

At the proxy-tuned peak LR of 1.6e-02, D2Z is 2.15% better than 10×. We also trained 93.75% sparse models with a variety of other decay ratios between 10× and D2Z, and present these results in Figures 21 and 22. Here we see a largely linear decrease in loss with a linear increase in the decay ratio (i.e., a linear decrease in the minimum LR). These are encouraging findings in the sense that

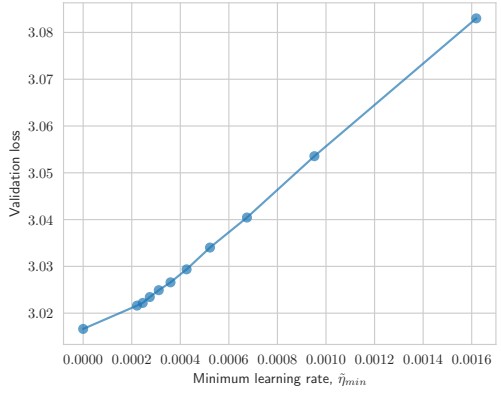

Figure 21: **Decay ratio sweep, part 1** (610M): Validation loss for models trained with SμPar, with 93.75% unstructured weight sparsity, as we vary the LR decay ratio. The x-axis plots the *minimum* LR, i.e., the LR at the final training step. As the ratio increases from 10× (right-most point), where the minimum LR is 10% of the proxy-tuned rate, to 0% (D2Z, left-most point), validation loss decreases roughly linearly.

Figure 22: **Decay ratio sweep, part 2** (610M): Validation loss for models trained with SμPar, with 93.75% unstructured weight sparsity, for a LR decay ratio of 10×, 100×, 1000×, and D2Z (using a different sparsity pattern than part 1). As in part 1, validation loss decreases roughly linearly with the minimum LR.

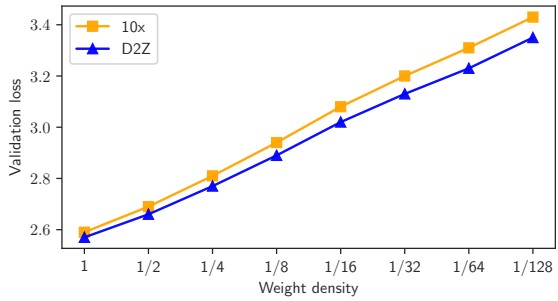

Figure 23: **Comparison of decay schedules when using *varying* weight sparsity** (610M): Validation losses for models trained with SμPar, as unstructured weight sparsity increases (density decreases). All models trained with the same number of training tokens (11752 steps, equivalent to 20 TPP for the fully-dense models). As sparsity increases, the number of trainable parameters decreases, and thus tokens-per-(non-zero)-parameter (and the benefit of D2Z) increases.

D2Z can seemingly be used directly on a range of problems, without having to worry about tuning a problem-specific LR decay ratio (e.g., 50× or 100×).

In Figure 23, we investigate the gap between D2Z and 10× at the proxy-tuned peak LR across different sparsity levels. Note that increasing sparsity effectively leads to a corresponding decrease in the number of trainable parameters. Since we use a fixed number of training tokens in each case, as the number of parameters decreases, the number of tokens-per-parameter (TPP) *increases*. In this sense, we note the relative differences between D2Z and 10× are consistent with our results in Figure 6 — as TPP (and gradient noise) increases, D2Z performs relatively better.

## C.9 ERROR BARS

***Finding 15****: Run-to-run variation in results is low.*

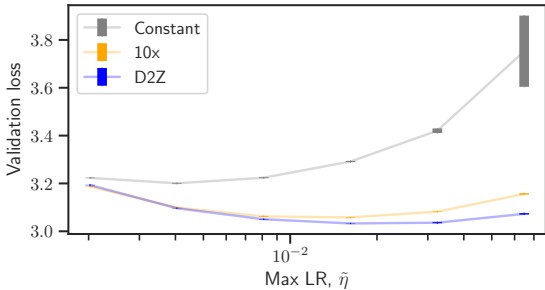

Figure 24: **Stability across random seeds** (111M, 20 TPP): Validation loss variance for different maximum LRs at different decay ratios. Each point corresponds to the mean validation loss over 5 separate training runs with different random seeds; the error bars give the standard deviation. Beyond the instabilities at high learning rates, run-to-run variance is remarkably low at this scale.

Taken as a whole, our results are remarkably stable: empirical results for different model scales, training durations, batch sizes, weight decays, and weight sparsity settings largely behave as predicted by theory. Since all validation runs are performed on 1.1B tokens, any significant run-to-run variance must arise during training. To quantify this variance, we repeated 111M-model 20 TPP training four additional times, resulting in 5 total validation loss results for each original training run. Figure 24 confirms that run-to-run variance is remarkably low. The only significant variance arises in *Constant* results at the highest learning rate. Here, the training loss sometimes spikes at various points during training, and the final validation loss can be significantly higher for models that cannot recover sufficiently following the spike.

### C.10   NANOGPT EXPERIMENTS

> ***Finding 16****: Improvement of D2Z over 10× persists in NanoGPT models — when numerical issues are solved.*

We also compared 10× versus D2Z by training NanoGPT models using the NanoGPT codebase (Karpathy, 2024). NanoGPT uses the standard parameterization. We configured these models to be largely similar to our 111M-parameter models, also using a weight decay of 0.1, a context length of 2048, and the GPT-2 vocab size of 50257. Key differences here are that we do not include bias weights, and we trained on the OpenWebText dataset (Gokaslan & Cohen, 2019). Experiments are run on Nvidia A10 GPUs.

As mentioned in Section 5, we initially tested NanoGPT in `bfloat16` precision. Here, at the default NanoGPT learning rate of 6e-4, 10× performed slightly better than D2Z. As we pushed the LR 50% higher (to 9e-4), both 10× and D2Z had higher loss, and by 1.2e-3, the loss from 10× doubled.

We suspected that numerical issues may be causing the instabilities, and repeated our experiments in `float32`. At this precision, we were able to successfully increase the LR, by factors of two, up to 32 times the default. At these levels, we do see the familiar gains of D2Z over 10× (Figure 25). We note there is nothing fundamentally limiting about `float16` precision itself - indeed all our main experiments were done using this precision. Rather, `float16` is simply problematic at a high learning rate in the NanoGPT codebase (see Gray et al. (2024, Appendix C.2) for a potential root cause of this instability).

These experiments demonstrate that a comparison between D2Z and 10× may serve as a kind of *diagnostic* of whether a model is being trained at optimal peak LRs: if a 100M+ model trained to 20 TPP does not see roughly 1% gains from using D2Z, it is likely the LR is not high enough. In order to raise the LR further, efforts to stabilize the model, perhaps including $\mu$P or other techniques (Wortsman et al., 2023), may be warranted.

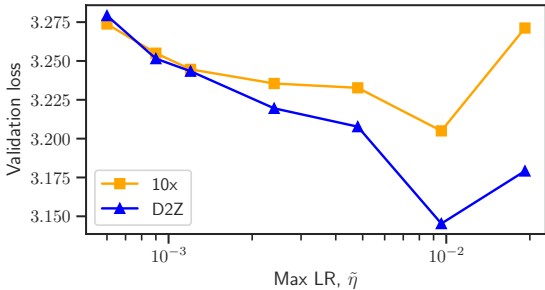

Figure 25: **NanoGPT results** (111M, 20 TPP): Validation loss for different LR and decay combinations, for a *NanoGPT model*. With `float16` precision, we were not able to train above 6e-04 without instabilities (first point in curves). Moreover, at $\tilde{\eta} = $ 6e-04, 10× performed better than D2Z. After switching to `float32`, we were able to train at higher $\tilde{\eta}$ values, where D2Z demonstrates its familiar superiority over 10×.

Table 6: **Model architecture and batch sizes for scaling law experiments**.

| Model | vocab. size | $d_{\text{model}}$ | $n_{\text{layers}}$ | $d_{\text{head}}$ | $d_{\text{ffn}}$ | batch size |
|---|---|---|---|---|---|---|
| 272M | 84992 | 1024 | 14 | 64 | 2912 | 128 |
| 653M | 84992 | 1536 | 18 | 128 | 4096 | 160 |
| 1.39B | 84992 | 2048 | 24 | 128 | 5472 | 208 |
| 2.75B | 84992 | 2560 | 32 | 128 | 6832 | 256 |

## C.11 SCALING LAW EXPERIMENTS

> **Finding 17**: *Improvement of D2Z over 10× grows across model scales, for a different model and training setup.*

Encouraged by the results of D2Z at smaller scales, we began testing D2Z in some of our frontier model efforts. Frontier models do not provide scope for hyperparameter tuning at scale. Thus it becomes important to derive scaling laws to forecast loss at larger scales, based on the loss with a sequence of smaller models.

For this set of experiments, we tested Llama-style (Touvron et al., 2023a) architectures, except using LayerNorm instead of RMSNorm, and multi-head attention instead of group-query attention. We use $\mu$P here as well, and a batch size scaling law to determine an optimal batch size for each model scale. These models also use ALiBi embeddings (Press et al., 2022) and SwiGLU (Shazeer, 2020). Here, the context length is 8192 tokens, and we use tied embeddings. We also re-tuned our $\mu$P proxy-model hyperparameters.

We used a bilingual data mix of English, Arabic and source code samples mixed in a 2:1:0.4 mix ratio. English data is from the Pile (Gao et al., 2020), Arabic uses a proprietary dataset (Sengupta et al., 2023), and the source code comes from the GitHub portion of the Pile.

To derive scaling laws to compare D2Z and 10× models, we used the power law functional form $y = cx^m$, where $x$ is the pre-training FLOPs, $y$ is the loss on the Pile validation set, and $c$ and $m$ are parameters to be fit. To fit these parameters, we transformed our Loss-to-FLOPs equation, $y = cx^m$, to a logarithmic form, $log(y) = log(c) + mlog(x)$, and fit the slope and intercept parameters of this line using a least-squares linear regression. We trained models at four sizes to compute-optimal 20 TPP (Table 6), and computed total FLOPs spent as well as validation loss on the Pile. We then fit the power law free parameters to obtain our scaling laws.

Encouragingly, here we find the scaling law slope of D2Z is roughly 2.5% better than 10× decay (Figure 26). This translates to an improvement of roughly 1% at 1.3B and 2.7B scales, broadly similar to our earlier results at 1.7B scale. Projecting our scaling law to a 70B model trained to a compute optimal 20 TPP, D2Z would achieve a roughly 2% loss improvement over 10× decay.

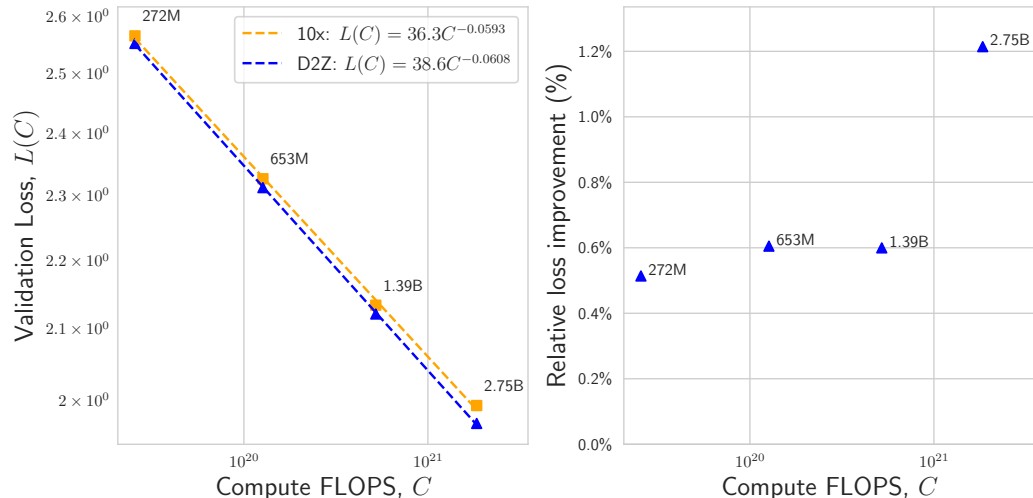

Figure 26: **Scaling law comparison of decay schedules**: Loss-to-FLOPS scaling law fits for models trained to 20 TPP (8192 context lengths, Llama architecture, Pile validation data). The power law fit for D2Z models has a steeper slope than the scaling law for $10\times$ models, implying a growing performance gap for larger model sizes.

## D    ADAMW AS CONVEX COMBINATION OF WEIGHT UPDATES, DRIVEN BY LR SCHEDULE: FURTHER DETAILS

We have begun using plots of the weight update coefficients internally to proactively analyze LR schedules. These plots give us insight into the timescale over which training data is integrated into the model parameters, and can be obtained without actually performing any training. Moreover, we can also use the extended EMA perspective in order to design novel LR schedules that have desirable blends of weight updates. Optimizing these coefficients thus provides a dual to optimizing the LR decay function, which is why we refer to these coefficients as the *dual coefficients* of the LR schedule. In this section, we first describe the design of one such LR schedule, and then we follow with example plots for this and other schedules.

### D.1    TRULY SCHEDULE-FREE LR SCHEDULES

Recall that, for a moving average with time-varying smoothing, $y_t = (1 - \alpha_t)y_{t-1} + \alpha_t x_t$, we use $c_{t,i} = \left(\prod_{j=i+1}^{t}(1 - \alpha_j)\right)\alpha_i$ to denote the dual coefficients, i.e., the contribution of input $x_i$ to output $y_t$ at time $t$. *Constant* schedules, or schedules with a long constant phase such as *WSD*, are not truly "schedule-free" because their peak LR setting affects the $(1 - \alpha_j)$ terms in the dual. Different LRs will correspond to different effective timescales over updates, and thus different LRs will be optimal for different training durations.[6]

In contrast, we now derive a schedule such that coefficients are always weighted equally, at every training step. First, it can be shown that:

$$\frac{c_{t,i+1}}{c_{t,i}} = \frac{\alpha_{i+1}}{(1 - \alpha_{i+1})\alpha_i}. \tag{7}$$

For the coefficients to be uniform at any step $t$, we require this ratio to be 1. Unity of this ratio implies that smoothing evolves $\alpha_{i+1} = \frac{\alpha_i}{(1+\alpha_i)}$. Assuming a fixed weight decay (so $\alpha_i = \eta_i \lambda$), coefficients will be equal if the LR evolves:

$$\eta_{i+1} = \frac{\eta_i}{(1 + \eta_i \lambda)} \tag{8}$$

---

[6]The different shapes of the coefficients for different LRs can be seen in Figure 28a for *Constant* and Figure 29d for *WSD*.

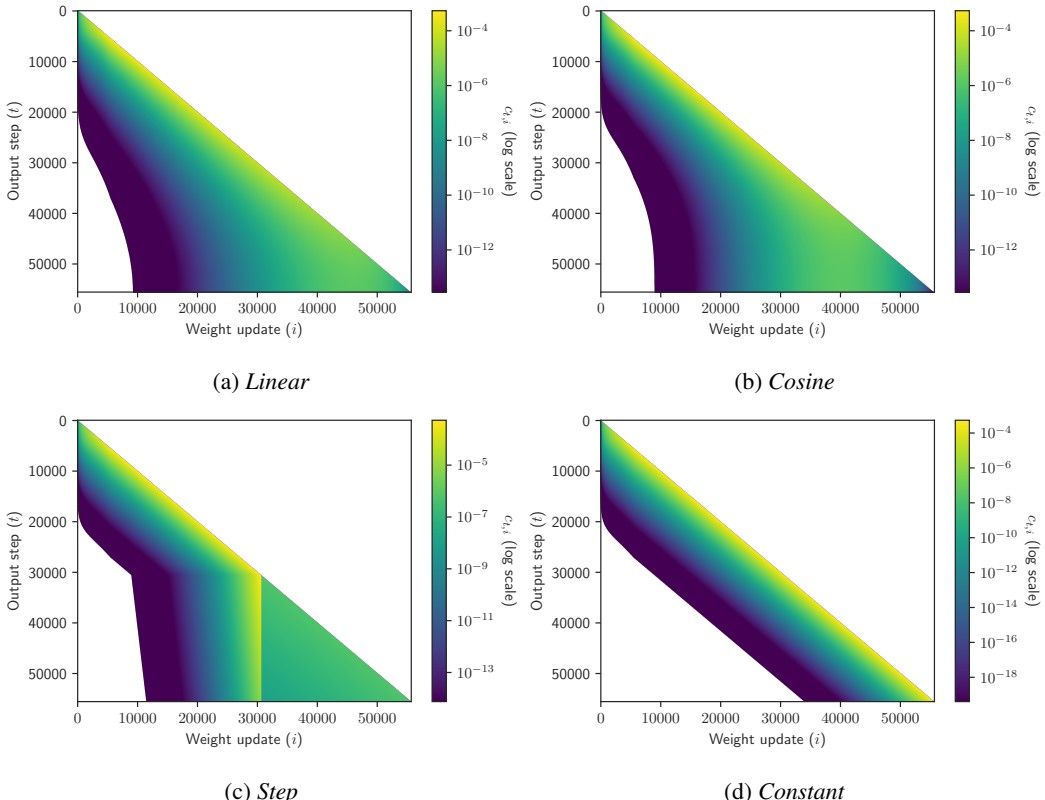

Figure 27: **EMA perspective: weight update contribution across steps**: Convex combination of weight updates, with color indicating value of combination coefficients $c_{t,i}$: each $c_{t,i}$ gives the contribution of the $i$th update (across x-axis) to model weights $\theta_t$ across steps $t$ (y-axis). Note that LR schedules and coefficients corresponding to the final step only were presented earlier in Figure 2 (except for *Constant*). Coefficients correspond to settings for 111M-parameter models trained to 200 TPP: $t$=55680, $\tilde{\eta}$=1.6e-02, $\rho$=$^1/_3$, $\lambda$=0.1.

We can initialize $\eta_0$ to some value and iterate Equation (8) to generate the full LR schedule. We call this the *Rational* schedule since it is both a rational expression of $\eta_i$ and a very reasonable approach: regardless of how long we train, all weight updates contribute equally. At each step we effectively decrease all prior coefficients such that they now equal the coefficient of the current update, $\alpha_t$.

With $\lambda$=1 and no warmup, this schedule evolves like $^1/_n$. However, in order to distance ourselves from the initial conditions, we can warmup the LR in the usual manner to the desired peak LR, then switch on rational decay, ensuring equal coefficients going forward. The full schedule is depicted in Figure 28d. However, such a schedule will almost surely not work as effectively as *Linear* D2Z for fixed-duration training, since it lacks the cooldown phase where gradient noise is minimized. Indeed, $^1/_n$ has performed relatively poorly in prior studies (Ge et al., 2019; Defazio et al., 2023). However, we introduce it here as a promising schedule for *continuous* pre-training.[7]

## D.2 LR CURVES AND DUAL COEFFICIENTS

In this section, we provide some extra figures that were referenced in the main paper. Figure 27 shows the dual coefficients at every step of training, using color to indicate the coefficient value (log-scale). Every horizontal row/step of Figure 27 reflects the coefficients at that step, essentially providing a version of Figure 2 but at each step. Figure 28 provides the LR schedules and dual coefficients for some of the schedules discussed in the main paper, as well as our proposed *Rational*

---

[7]In fact, $^1/_n$ decay has long been regarded as optimal for strongly-convex loss (Robbins & Monro, 1951), and optimal in other contexts when combined with *averaging* (Defazio et al., 2023). Since averaging is an alternative to cooldown (Hägele et al., 2024), *Rational* plus averaging is a compelling schedule-free direction.

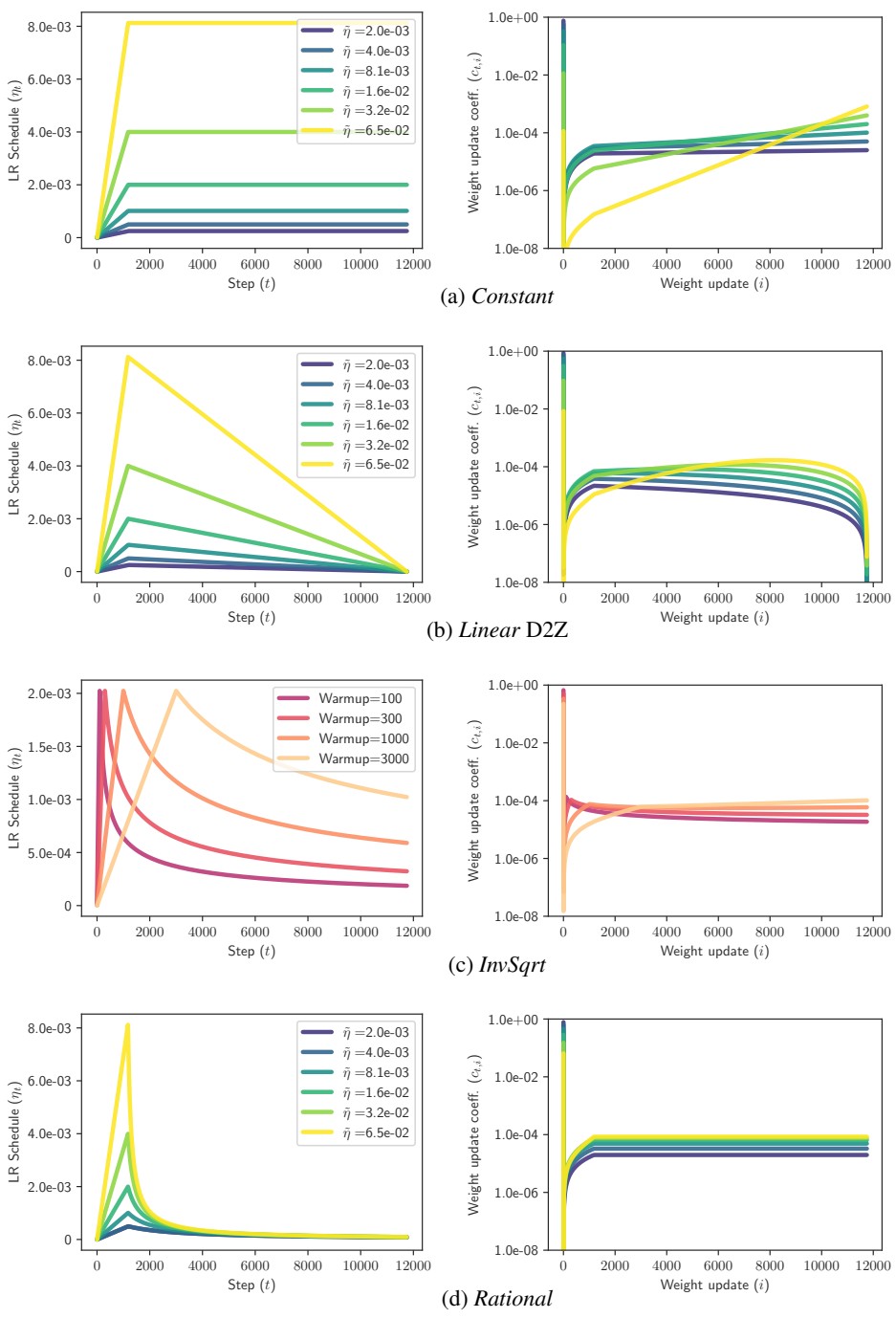

Figure 28: **LR curves and dual coefficients for various common LR schedules** as well as the proposed *Rational* approach (Appendix D.1): Dual coefficients shown at final training step (right side) for 610M-parameter, 20 TPP training ($t$=11752, $\rho$=1/8, $\lambda$=0.1). For *InvSqrt*, we vary the warmup and fix $\tilde{\eta}$=1.6e-02 for all curves. Note coefficient values are plotted on log-scale

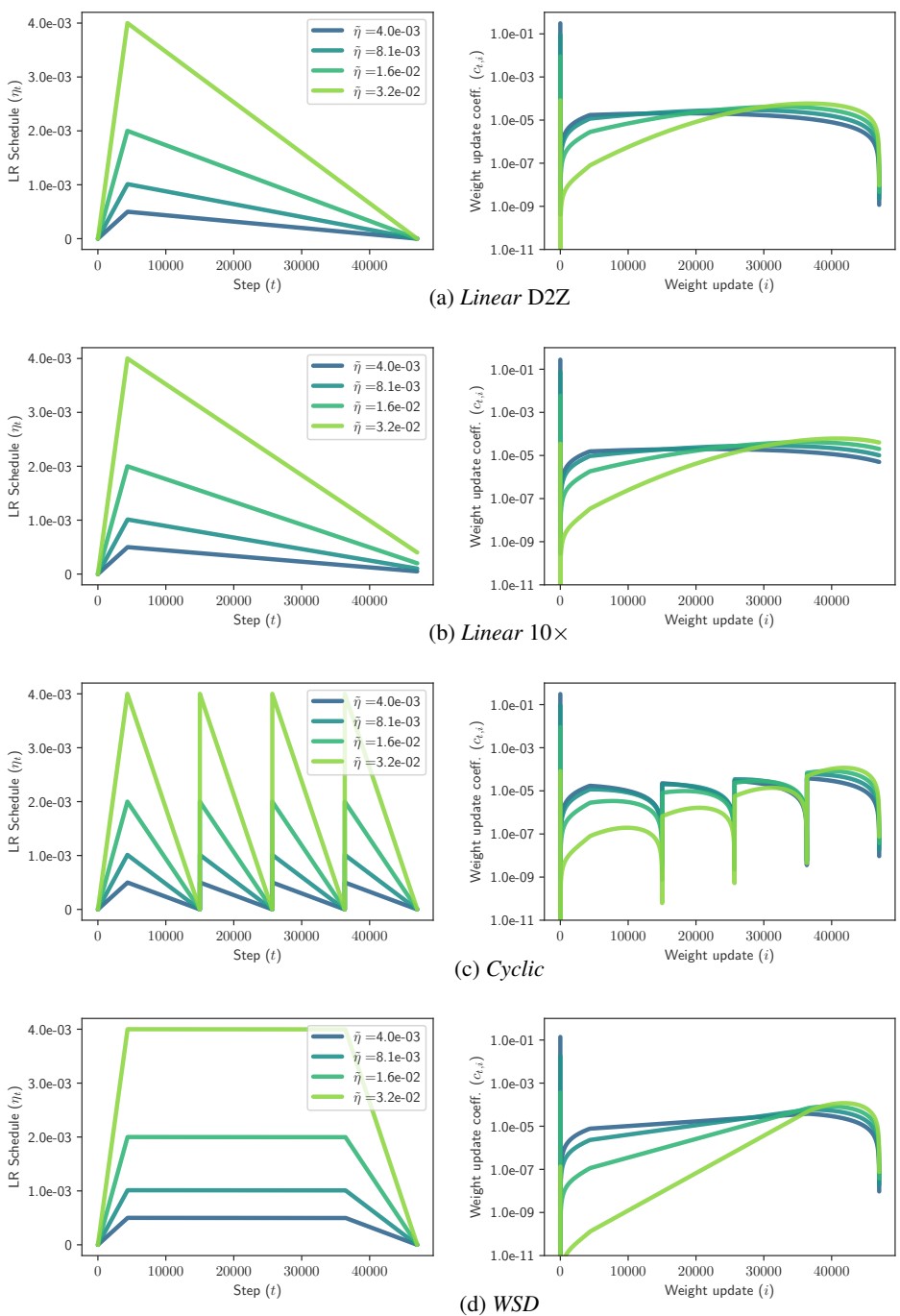

Figure 29: **LR curves and dual coefficients for standard vs. *continuous* schedules**: Comparison of standard *Linear* schedules versus *Cyclic* and *WSD*. On the left are the exact LR schedules used in Figure 8 evaluations, i.e., 610M-parameter, 80 TPP training ($\rho$=1/8, $\lambda$=0.1). Dual coefficients shown at at final training step $t$=47008 (right side). Note coefficient values are plotted on log-scale.

schedule, which combines all the prior weight updates (after warmup) equally at every step. Finally, Figure 29 gives the LR schedules and dual coefficients for the comparison to *WSD* and *Cyclic* in Figure 8.

It is worth re-iterating that the dual coefficients can be computed separately from any actual training. They are mathematically equivalent to the LR schedule itself and simply provide a perspective on how the weight updates combine to form parameters, when using the AdamW optimizer. Furthermore, it is also worth noting the vertical bar at step 1 in the dual coefficient plots; this bar reflects the coefficient on the initial, random weights. To some extent, this $c_{1,t}$ value can serve as an indicator of how far the model has moved from initial conditions, i.e., the extent to which the model has reduced the bias. In general, if $c_{1,t}$ is too high (e.g., when it outweighs the sum of the other coefficients), then bias is likely significantly hindering learning. Two effective ways to reduce $c_{1,t}$ and thus reduce bias are to (1) raise the peak LR, and (2) train for more TPP. In contrast to these two methods, raising weight decay, $\lambda$, can also decrease $c_{1,t}$, but is counterproductive for reducing bias because it also reduces the scale of weight updates, as noted in Hypothesis 4. Likewise, using smaller batches also reduces $c_{1,t}$, but is likewise counterproductive if the batches become too small, to the extent that gradient noise increases. However, D2Z is more robust to such noise than $10\times$ or *Constant* decay. Further investigating the interplay of $\lambda$, batch size, and peak learning rate is important future work.

