# OpenReview forum: "Straight to Zero: Why Linearly Decaying the Learning Rate to Zero Works Best for LLMs"
_ICLR.cc/2025/Conference — ICLR 2025 Poster_

### Official Review · Reviewer_f1Pq · 2024-10-31

**Soundness:** 3
**Presentation:** 4
**Contribution:** 3
**Rating:** 8
**Confidence:** 3

**Summary:**

The submission studies learning rate scheduling for LLM pre-training, in particular, the paper notices that, in high TPP setting, i.e. when the model is trained much longer than what scaling laws recommend, a linear decay to zero (D2Z) scheduling gives optimal performance and converges faster than the commonly used cosine decay to 0.1 max_lr. To be more specific, the method utilizes a recently proposed interpretation AdamW, which suggests that the model weights at a certain time step is an exponential moving average of past updates, with the smoothing factor determined by learning_rate * weight_decay_coefficient. Based on this observation, the authors suggest that linear decaying to zero allows for more updates to be averaged, therefore reducing the variance, which is a crucial component in the later phase of the optimization (which is the pattern for high TPP setting).

The paper then empirically verifies the claims on an **extensive** set of experiment settings, all confirming the benefit of linear D2Z: Including better validation loss in high TPP setting, less sensitivity to the choice of batch size, learning rate setting. Additional experiment results are also presented in the appendix.

Overall, I find the paper well written, the presentation is very clear, and the empirical evidence convincing.

**Strengths:**

- The paper studies an important problem: How to schedule the learning rate, a critical but under-studied issue, until recently by Defazio et al. (2023).

- The paper also studies the role of weight decay (as well as its interaction with learning rate and learning rate scheduling), which is also under studied, until recently by Andriushchenko et al. (2023) and  Wang & Aitchison, (2024)

- The argument, despite not being supported by strict proof, is built upon classic optimization theory (bias-variance decomposition) and simple linear algebra (EMA perspective of AdamW), so I find the claims fairly rigorous.

- The authors explore a wide range of confounding factors that could affect the conclusion on "best learning rate schedule", such as peak learning rate, batch size, weight decay, training length, and observe similar pattern across all settings, therefore I find the conclusion very solid and would consider starting using linear D2Z in my own LLM experiments.

**Weaknesses:**

- I think there is room for improvement in the writing: Section 3.1 and 3.2: The paper's main argument is about the optimality of linear D2Z, but these two sections do not talk about, e.g. why cosine decay is suboptimal (which was only briefly mentioned in the caption of FIg.2) or why decay to zero is better than decay to 0.1 max_lr. Although I do understand the connection between the suboptimality of other scheduling and the reducing variance perspective, more explicit arguments can be made to improve the readability.

- The cyclic learning rate scheduling (or is it just cosine learning rate decay?) is used in experiments as a baseline but not explained or discussed in, e.g. Figure. 2.

- It would be nice if more details about the computational resources, model configuration (e.g. number of self attention layer, head, hidden dimension size) can be revealed.

- It would be nice if the experiment code can be released.

**Questions:**

- I really like the author's argument on why, given a fixed value of eta * lambda, modifying eta does not have the same effect as modifying lambda, since having a lambda too large would affect the bias reduction. I wonder if the authors have considered mitigating this issue by modifying the weight initialization scale: The EMA perspective suggests that the weight will end up having a magnitude ~ 1 / lambda, and if lambda is large, the relative difference (bias) between the final weights and initialization of fixed magnitude will indeed be very small, however, one can consider additionally scale the initialization by 1 / lambda, such that the bias could potentially be reduced at larger lambda. Such a strategy is also suggested by Wang & Aitchison, (2024) in Appendix A, Eq. 21 (but under fully scale invariant network assumption).

- I wonder what does the author think of the reason behind Finding 4, i.e. weight decay is beneficial only used together with certain learning rate scheduling. Such behavior is also observed when training ResNet on CIFAR-10, Loshchilov & Hutter, 2017 noticed that weight decay can improve test accuracy only combined with learning rate scheduling.

- Notice that the dataset scaling rule Wang & Aitchison (2024), also implied that the learning rate should increase as batch size increase, in particular, the authors suggest that the key hyperparameter is M * eta * lambda, where M = number_of_training_point / batch_size, therefore as batch_size increase, eta * lambda should also increase. But Wang & Aitchison (2024) mainly considered high-epoch settings with the dataset being repeated multiple times (e.g. standard CIFAR-10 training pipeline) so their claims may not be transferrable to LLM pre-training where the same training data point almost never show up twice.

- In a very extreme and unrealistic setting, where we can afford full batch training, in which case I wonder if weight decay can be entirely removed since there will be no gradient noise at all?

- I believe such interpretation can also be applied to nonadaptive optimizers, e.g. standard SGD or Sign-GD with momentum, would the authors think the claims still hold true?

- (Minor) Why is the batch size chosen as multiples of 63? This seems to be a weird number.

---

> ### Author Response · Authors · 2024-11-15
> **Response to Reviewer f1Pq (1/2)**
>
> Thank you so much for your detailed and valuable review, and for generously noting the paper’s "convincing" experiments and "very solid" conclusion.
>
> ### Regarding "when the model is trained much longer than what scaling laws recommend ... D2Z scheduling gives optimal performance"
>
> Just to clarify: in all cases where we train exactly to the compute-optimal frontier (as recommended by scaling laws), D2Z *does* improve over 10x (e.g., Figure 11), and yet you are correct that when training beyond this level, the gains increase further.
>
> ### Regarding "there is room for improvement in the writing: Section 3.1 and 3.2"
>
> This is a very fair point, and we appreciate your help here.  We will definitely improve these sections for the camera paper.  We will explicitly develop the idea that decaying the LR to zero allows for more updates to be averaged, reducing the gradient variance; and we will note that variance reduction grows in importance in the later stages of optimization, particularly for higher-TPP training.  We make points along these lines in Section 5.1, but adding discussion here will make the methods section much clearer.  We will also motivate the differences between cosine D2Z and linear D2Z from a loss surface perspective, as suggested by Reviewer sx5Y, and then subsequently show the connection to the EMA coefficients; this should also improve the presentation here.
>
> ### Regarding "the cyclic learning rate scheduling ... is used in experiments as a baseline but not explained or discussed"
>
> Yes, we only briefly mention in Section 4.2 that "Appendix Figure 27 provides full LR curves and dual coefficients for all models in Figure 7", and this includes the Cyclic LR schedule (third row of Figure 27).  We will add a separate figure that only shows the LR schedules (including Constant, Linear, Cosine, WSD, and Cyclic), and refer to this figure early in the main body of the paper, rather than only showing these schedules on the final page of the appendix.  Good point.
>
> ### Regarding "it would be nice if more details about the computational resources, model configuration ... can be revealed."
>
> True, right now we only say, in Section 5.1, that "Appendix A has full experimental details."  But we will revise this to note that "Appendix A has further experimental details, including details on the models (Appendix Table 2 providing the model width, number of layers, number of attention heads/head dimensionality, batch size, etc.), as well as details on the computational resources used during training."  We will also expand Table 2 to explicitly provide the number of heads in the multi-head attention ($=d_{model}/d_{head}$).  Please let us know if there's anything else that would be useful to include here.
>
> ### Regarding "it would be nice if the experiment code can be released"
>
> There is precedent in our organization for releasing code to accompany published papers.  In particular, we should be able to share the code for calculating and visualizing the EMA coefficients for arbitrary LR schedules, and for reproducing our results with the NanoGPT codebase.
>
> ### Regarding "modifying the weight initialization scale ... by 1 / lambda, such that the bias could potentially be reduced at larger lambda"
>
> This is an interesting suggestion.  Our overall philosophy has been to adopt the maximal update parameterization, where weights are initialized in order to ensure invariance to changes in model width.  From this perspective, the ability of weight decay to enable variance reduction later in training, without impacting bias reduction, can perhaps be viewed as a feature rather than a bug – it gives us a tool to specifically impact the later stages of optimization.  But this idea definitely merits further study, and the link to Wang & Aitchison’s Eq. 21 is worth noting in the paper, thanks!
>
> ### Regarding "[why] weight decay is beneficial only used together with certain learning rate scheduling"
>
> Thank you for the pointer to the prior work in Loshchilov & Hutter, 2017.  As discussed in Section 5.1, we believe that without LR decay, a constant LR is averaging over too few updates later in training.  Raising weight decay only further shortens the timescale over updates, so is not beneficial (although it might have been beneficial if much smaller constant LRs had been used).  Meanwhile, with D2Z, we can use weight decay to fine-tune the average timescale during the decay period.  This is more effective than adjusting the peak LR itself, as raising it too high causes instabilities, and too low affects movement from initial conditions.  10x decay (at an optimal LR) only mildly benefits from weight decay because it is somewhere in between these extremes.  We would like to pursue further experiments along these lines in follow-up work, so any thoughts that you have on this topic would be very welcome!  We will also revise this part of 5.1 to be more clear, and provide a forward-pointer to this discussion under Finding 4.

---

> > ### Author Response · Authors · 2024-11-15
> > **Response to Reviewer f1Pq (2/2)**
> >
> > ### Regarding "the dataset scaling rule ... also implying that the learning rate should increase as batch size increases"
> >
> > Yes, this is a good point: we will note that LR scaling with batch size is anticipated by the dataset scaling rule in Wang & Aitchison (2024).  We will expand on this under Finding 3 in our discussion of the batch size results in Figure 5, and in our discussion section.  Actually, in very recent work, we have found that while the LR does not scale perfectly linearly with batch size when using D2Z, the weight decay *does* do so (and, moreover, leads to lower loss!).  This again illustrates the targeted role of weight decay in the reduction of variance (as opposed to the LR's interaction with bias).  And this also shows the value of our contribution to the EMA perspective.  We will add some of these results to the paper in order to flesh out this discussion further.
> >
> > ### Regarding "full batch training ... can weight decay be entirely removed since there will be no gradient noise at all?"
> >
> > In related work, it had previously been hypothesized that the reason Adam succeeds over SGD in language modeling is because Adam better handles heavy-tailed gradient noise.  However, recent work by [Kunstner et al. (2023)](https://arxiv.org/abs/2304.13960) showed that even with deterministic, full-batch training, Adam still prevails.  It would be interesting to compare AdamW to Adam in the same setting.  Of course, we know from theory that for well-behaved optimization problems, convergence can be guaranteed without *LR* decay in the setting without gradient noise.  (This relates to our Equation (1) and is noted in [Bottou et al. (2018)](https://arxiv.org/abs/1606.04838) as "a standard result for the full gradient method with a sufficiently small positive stepsize").  We actually begin to see this in Figure 14, where the difference between 10x and D2Z shrinks as the batch size increases, but where we take the same number of overall *steps*.  Here, the optimal max LR increases (especially for 10x decay) as the batch size increases.  From the EMA perspective, this suggests the optimal weight decay would also increase, at least until we are past the critical batch size where gradient noise rapidly decreases.  So, these results suggest we would need higher weight decay as gradient noise decreases, at least when using AdamW.  Either way, this is definitely well worth studying further, and we will add some further interpretation along these lines to our very short discussion in Appendix B.5.  Thanks for raising this!
> >
> > ### Regarding "nonadaptive optimizers, e.g. standard SGD or Sign-GD with momentum, would the authors think the claims still hold true?"
> >
> > Yes, we would tentatively hypothesize that linear D2Z would be the optimal LR schedule for compute-optimal training of LLMs using non-adaptive optimizers.  The bias-variance perspective (Equation (1)) applies to non-adaptive optimizers, and we hypothesize compute-optimal TPP always leads to gradient noise dominating training, and thus necessitating maximal LR annealing.  At the same time, in our experiments where we set weight decay to zero, we found D2Z had only a minor advantage over 10x decay, so it’s certainly possible that 10x and D2Z may be quite close for other optimizers when disabling weight decay (let alone removing preconditioned gradients).  For any optimizer, claiming one LR schedule is "best" requires controlling for a number of confounding factors.  Unfortunately, controlling for these factors at scale is very compute-intensive, which is why it’s valuable when papers like ours do such "dirty work" for the community.
> >
> > ### Regarding "why is the batch size chosen as multiples of 63?"
> >
> > We use scaling laws to estimate the optimal batch size for different model scales, along the lines of those used in [Bi et al., (2024)](https://arxiv.org/abs/2401.02954).  These laws advised use of a batch size of 504 for the 617M scale.  From this base batch size, we swept batch sizes up and down by factors of two, revealing the common divisor of 63.  In the camera paper, we will revise to provide more details on how the initial batch sizes were selected.

---

> ### Comment · Reviewer_f1Pq · 2024-11-15
> **Response to author rebuttal**
>
> Thanks for addressing the comments!
>
> > Our overall philosophy has been to adopt the maximal update parameterization, where weights are initialized in order to ensure invariance to changes in model width.
>
> I understand this part, I am suggesting having a weight-decay dependent scaling to the initialization together with the 1 / sqrt{fan_in} MuP scaling, which I believe would not break MuP. But yes, this would beyond the scope of the current submission.
>
> >  Actually, in very recent work, we have found that while the LR does not scale perfectly linearly with batch size when using D2Z, the weight decay does do so (and, moreover, leads to lower loss!).
>
> Interesting finding!
>
> I have increased my score from 6 to 8. I recommend acceptance for this paper. Please improve the writing and presentation of Sec 3 accordingly, that will hugely improve the readability and easiness of understanding for the proposed idea.
>
> However, I have to admit that I only know about academia-size deep learning optimization problems but I have **zero** expertise and experience in LLM scaling law-related topics, therefore whether or not the paper's experiments are "small scale" goes beyond my scope of knowledge, I am open to other reviewer's comments.

---

> > ### Author Response · Authors · 2024-11-30
> >
> > Thanks so much for engaging in the discussion and for your support of the paper!  If at all possible, we would really value any feedback you can provide on the proposed revision of Section 3, posted as a comment above.  In our view, being more explicit about the experimental hypotheses and how they arise from our conceptual model already significantly improves the paper's readability.  We greatly appreciate the help you provided here.

---

### Official Review · Reviewer_3io8 · 2024-11-01

**Soundness:** 2
**Presentation:** 3
**Contribution:** 2
**Rating:** 5
**Confidence:** 4

**Summary:**

This manuscript empirically demonstrates that, with a fixed maximum learning rate, a simple linear decay-to-zero (D2Z) schedule often outperforms the widely-used cosine decay to 10% of the maximum, as well as other popular learning-rate schedules. It also shows that linear D2Z is robust to variations in maximum learning rate, weight decay, dataset size, and batch size.

**Strengths:**

The manuscript presents extensive experiments that validate the effectiveness of D2Z for training GPT-like models with 111M, 617M, and 1.7B parameters. It effectively illustrates D2Z's robustness across different maximum learning rates, weight decay values, dataset sizes, and batch sizes.

**Weaknesses:**

- Linear D2Z appears to function more as an engineering trick than a comprehensive algorithm. While it demonstrates empirical effectiveness, the theoretical justification is somewhat unconvincing. The authors attempt to explain its success through theoretical convergence analysis (Eq. (1)) and update rules involving weight decay (Eq. (3), Eq. (4)), but a unified framework to interpret the results clearly is lacking. It seems the authors selectively apply certain perspectives to support their experimental findings. Furthermore, the convergence bound in Eq. (1) applies only to convex optimization problems with SGD, which are significantly different from the training of large language models (LLMs) using Adam. In summary, a more rigorous mathematical framework that connects the empirical results to theoretical principles is needed.

- Although the manuscript presents simple scaling law experiments in Appendix B.10, the experimental details are unclear. The authors are encouraged to provide more comprehensive information on scaling law experiments for D2Z to demonstrate its applicability to larger models.

- D2Z requires prior knowledge of the total number of iterations, which is not ideal for continuous pretraining scenarios. In contrast, WSD and cyclic schedules  are  more effective for continuous pretraining.

**Questions:**

Subsection 3.3 has a weak connection to the overall manuscript.  Can you further clarify its relevance to the main findings about D2Z.

---

> ### Author Response · Authors · 2024-11-15
> **Response to Reviewer 3io8 (1/2)**
>
> Thank you very much for your thoughtful and constructive feedback, and for noting the extensive and effective experiments.  We are also grateful for all the reviewers scoring the paper highly in terms of presentation quality.
>
> ### Regarding "a more rigorous mathematical framework that connects the empirical results to theoretical principles is needed"
>
> We take your point regarding the lack of analytical results; there is always a tradeoff between the ability to derive formal theorems and the generality of the theory (i.e., requiring fewer assumptions).  What attracted us to the EMA perspective is that it applies whenever AdamW is used as the optimizer.  We view our work as a step forward in terms of the mathematics, as Wang & Aitchison’s prior work did not properly account for a dynamic LR schedule.  Moreover, Equation (1), defined for SGD with convex loss as you note, is still useful as intuition about training having a phase where we move away from initial conditions, followed by a phase where we must primarily deal with gradient noise.  Here, another contribution of our work is to note the EMA perspective does also recognize different phases.  Indeed, we can quantify the dependence on the initial weights (bias), and we can show analytically how this is reduced via a larger LR, but not reduced via a larger weight decay.  This is the point noted by reviewer f1Pq: "I really like the author's argument on why, given a fixed value of eta * lambda, modifying eta does not have the same effect as modifying lambda, since having a lambda too large would affect the bias reduction."  This contribution reconciles the results in Figure 6, which otherwise are inconsistent with the original interpretation in Wang & Aitchison.  Moreover, as we will note in the camera version of the paper, this contribution also suggests that for training to a higher tokens-per-parameter (training with a greater proportion of gradient noise), it may be more effective to reduce weight decay than to reduce the LR.  This is a timely suggestion given that papers continue to appear that observe and recommend purely a decrease in the max LR for high-TPP training (e.g., [Bjorck et al., 2024](https://arxiv.org/abs/2409.19913)), which we view as counterproductive.
>
> That being said, we agree with Reviewer f1Pq’s point that the presentation in 3.1 and 3.2 can be improved, and unified around the concept of variance reduction.  We will revise these sections to explicitly describe why D2Z might improve over 10x decay from a variance reduction perspective (this is mainly discussed now in Section 5.1).  We also believe motivating the differences between cosine D2Z and linear D2Z from a loss surface perspective, as suggested by Reviewer sx5Y, and then subsequently showing the connection to the EMA coefficients, will greatly improve readability here as well.  Moreover, we will make clear where the theory is still lacking, and we will note our hope that, going forward, further theoretical principles can be developed to further explain our extensive findings.  Thanks for your suggestions here.
>
> ### Regarding "more comprehensive information on scaling law experiments"
>
> This is a great point.  First of all, as we noted in response to Reviewer sx5Y, we will revise the main paper to discuss these experiments (and our sweeping of max LR for 1.7B-parameter models), rather than only discussing these results in the appendices.  In terms of further experimental information:
> - While we provided information on model architecture for these experiments in Table 6, we neglected to include the batch size.  We will do so for the camera-ready paper.
> - Moreover, we did not include a table along the lines of Table 3, documenting the specific LR schedule (warmup, cooldown, and total tokens).  We will do so for the camera-ready paper.
> - We will also provide further details on our dataset mix, and describe in further detail the proprietary Arabic dataset that was partly used in the training of these models.  For our submission, we suppressed a citation to this data in order to maintain anonymity.
> - We will provide further details on how we obtained our power law fit.  Essentially, we transformed our Loss-to-FLOPs equation $y=cx^m$ to a logarithmic form, $log(y)=log(c) + m log(x)$, and fit the slope and intercept parameters of this line using a least squares regression.
> - Importantly, we will also provide the fitted power law exponents (-0.0608 for D2Z, -0.0593 for 10x decay), which strongly indicate that the gains from D2Z will continue to improve as we scale up models.
> - Finally, we will add a subplot to Figure 24 to explicitly show the relative loss improvement of D2Z over 10x decay (from 0.51% at 256M, to 0.6% at 590/1.3B, to 1.21% at 2.3B).
>
> If there are other specific details that you would like to see in this section, please let us know.

---

> > ### Author Response · Authors · 2024-11-15
> > **Response to Reviewer 3io8 (2/2)**
> >
> > ### Regarding the fact "D2Z requires prior knowledge of the total number of iterations"
> >
> > Just to be clear: this is a *limitation* of the approach, but not a *weakness* of the paper, correct?  Indeed, many LLMs are being trained for a fixed number of training tokens (with compute and data resources budgeted carefully in advance of training), so we believe determining the optimal schedule for the fixed-duration scenario is very well motivated.  That being said, we should have mentioned this explicitly as a limitation in Section 5.3, and will do so in the camera copy.  Of course, this limitation is mitigated by the fact there are now effective strategies for continuous pre-training that can re-warm the LR (and use data re-play) in order to continue training from a D2Z checkpoint (e.g., [Ibrahim et al., 2024](https://arxiv.org/abs/2403.08763)).
> >
> > ### Regarding the relevance of Subsection 3.3
> >
> > It is true that this section is perhaps less relevant to the main findings about D2Z.  However, we also did not do a good enough job in establishing the connection, so we appreciate that you raised this issue.  The motivation for this section relates to the previous point about continuous pre-training: if the rationale for avoiding D2Z is that it is not appropriate for continuous pre-training, does that mean that other approaches like a constant LR, or WSD, or an inverse square root schedule are to be preferred?  The key point of this section is really the first paragraph: even constant/WSD schedules have an implicit dependence between:
> > 1. the timescale over which weight updates are averaged, and
> > 2. the maximum LR.
> >
> > This dependence leads to a different maximum LR being optimal depending on the (unknown-in-advance) size of the dataset, a dependence that we show for a Constant schedule in Figure 3, and which has also been observed in prior work, e.g., in [Shen et al. (2024b)](https://arxiv.org/abs/2408.13359).  In this sense, constant schedules should not be preferred to D2Z for continuous pretraining on this basis alone.  The remainder of this section is just to illustrate that with the dual view of the LR schedule, we can design a schedule whose timescale is truly independent of the maximum LR, but perhaps the mathematical details of this idea are best left to the appendices.

---

> ### Comment · Reviewer_3io8 · 2024-11-28
>
> Based on the current response, I will maintain my rating of 5. From my perspective, Linear D2Z functions more as an engineering heuristic than a comprehensive algorithm. The authors mentioned in their response that they would provide additional theoretical analyses to establish its foundations, and I was waiting these in the revised manuscript. However, until now I have not observed any modifications addressing this aspect. Therefore, I will keep my rating unchanged.

---

> > ### Author Response · Authors · 2024-11-30
> >
> > Thank you for your continued engagement and helpful feedback!
> >
> > We do appreciate there is a difference between simple evaluation of an "engineering trick", and a systematic, hypothesis-driven approach. In response to your points, we've significantly revised Section 3 to clarify our conceptual model, which underpins the experimental hypotheses. This model predicts the patterns observed in our results, guiding practical LR schedule decisions for LLM training.
> >
> > The revised Section 3 (added as a comment above) now more explicitly links our conceptual model to the experimental hypotheses, organizing the discussion around how both bias and variance are influenced by learning rate and weight decay adjustments.  In particular, as we noted in our original rebuttal, the model predicts that it may be more effective to reduce weight decay than to reduce LR when training at high TPP.  We ran some additional experiments to test this, and found it to hold.  We will include these additional results in the camera-ready paper.
> >
> > E.g., at 200 TPP, for a 111M-parameter model:
> >
> > |     Tuning approach    |     Tuned 10x Decay hyperparameters    |     10x Decay Loss    |     Tuned D2Z hyperparameters    |     D2Z Loss      |
> > |------------------------|----------------------------------------|------------------|----------------------------------------|--------------|
> > |     Default LR/WD      |     $\eta$=1.6e-02, $\lambda$=0.1      |     2.890        |     $\eta$=1.6e-02, $\lambda$=0.1      |     2.810    |
> > |     Tuning LR only     |     $\eta$=0.4e-02, $\lambda$=0.1      |     2.853        |     $\eta$=0.8e-02, $\lambda$=0.1      |     2.808    |
> > |     Tuning WD only     |     $\eta$=1.6e-02, $\lambda$=0.001    |     **2.835**        |     $\eta$=1.6e-02, $\lambda$=0.05     |     **2.805**    |
> >
> > We acknowledge the critique regarding the absence of a unifying mathematical formalism. The complexity of LLM training dynamics makes purely theoretical approaches challenging. Our empirical, model-based approach provides a functional framework that aids in effectively navigating LR schedules.
> >
> > Your feedback has been instrumental in enhancing our paper, and we would value any thoughts you have on the improved Section 3.

---

### Official Review · Reviewer_sx5Y · 2024-11-05

**Soundness:** 2
**Presentation:** 3
**Contribution:** 2
**Rating:** 6
**Confidence:** 4

**Summary:**

This paper analyses the role the learning rate scheduler for LLM pre-training. The authors mainly focus on two questions:
- can we replace the widely used cosine learning rate scheduler by a simpler linear decay?
- can we do better than the commonly used rule to take the final lr as 1/10 of the peak lr?
The authors answer positively those two questions, demonstrating that a linear lr decaying to 0 works best for small models (most experiments are on 600m models, a few on 1.7B models) on the slimpajama dataset.
The authors conduct many ablations on the subject, including training length, sensitivity to peak lr, and role of weight decay.
The authors also propose a theoretical framework to try to understand this, based on unrolling EMAs with varying step-sizes.

**Strengths:**

- this paper studies a very important problem, that of the training dynamics of LLMs
- it is well-written and easy to read
- it makes a compelling story
- the paper convincingly demonstrates the importance of completely annealing the learning rate for small models on one dataset

**Weaknesses:**

- The main weakness of this work is the small scale of the experiments. For instance, the base learning rates for the small-scale models used in this paper are very high (cf table 2 and fig.10, best lr is around 3e-3), so decaying it to 10x still yields a high LR at the end of training. For bigger models, the base lr is much lower (typically 3e-4 or 1e-4 for 7B models, even lower for larger models). Therefore, the final lr when using the 10x technique is low in this case. Hence, it is very likely that the impact of d2z would be far less in these cases (for instance, in Fig. 10, there is barely any difference between d2z and 10x for these learning rates).
In order to address this concern properly, this paper needs to show that the benefits of d2z do not completely vanish at large scales. A similar base lr sweep to that conducted for the 600M models for the 1.7B model would benefit the paper greatly.


- The methods section seems very vague:
    - There are no formal statement, everything is hand-wavy
    - The main problem with the discussion regarding the EMAs, and summarizing everything with the variables $c_{t, i}$, is that it makes it seem that the variables $x_t$ in the parameter’s update EMA (eq.3) are external variables to the problem, while clearly they depend on the parameters $y_t$ themselves. This makes all the discussion in section 3 flawed.

- The impact of the dataset is another thing that is not discussed, while it will change the learning curves. This warrants at least a discussion, clarifying that the phenomenon observed here might not happen for datasets of different sources/quality.

[Edit after discussion, see replies in the forum: score 3-> 6]

**Questions:**

- what base lr is used to train the 1.7B models?
- It is doubtful that d2z is optimal: with d2z, the last iterations have an almost zero learning rate, yielding no progress and effectively wasting these last steps. This effect is worsened for the cosine schedule, which spends more time at low lr's. Having a non-zero value for the last lr would be better: how about a decay to, e.g., 100x? I understand that this leads to another hyperparameter to tune, which is cumbersome, but this could also be discussed.

---

> ### Author Response · Authors · 2024-11-15
> **Response to Reviewer sx5Y (1/2)**
>
> Thank you for your very helpful review, and your kind words regarding the importance of the problem, the well-written and compelling story, and the convincing experiments on the SlimPajama dataset.  We will address your main concerns by doing a better job of describing the larger-scale experiments and additional datasets in the main body of the paper (rather than only in the appendices).  We will also provide additional results at the 1.7B scale.
>
> ### Regarding "A base lr sweep … for the 1.7B model would benefit the paper greatly"
>
> This is an important point.  In fact, this sweep is shown in the appendix in Figure 11, right-hand-side.  The figure shows that even when each LR schedule uses its optimal peak LR, the relative loss improvement of D2Z over 10x decay actually *increases* at 1.7B scale (from 0.8% at 617M to 1.2% at 1.7B).  While the main paper focused on 617M models (where we could afford to perform more ablations), we unfortunately neglected to mention sweeping the base LR for some of our largest/most-expensive models (training 12 different 1.7B LLMs for this figure, each on 34.3B tokens!).
>
> ### Regarding "the base learning rates for the small-scale models used in this paper are very high"
>
> Note that MUP ([Yang et al., 2022](https://arxiv.org/abs/2203.03466)) is a technique that allows use of the same base LR (before adjustment) at different model scales.  For all figures that sweep LR for MUP models, we provide this base LR, $\tilde{\eta}$, on the x-axis.  The actual LR, $\eta$, is scaled down by a factor of 3 for 111M models ($d_{model} = 768$), and by 8 for the 617M and 1.7B models (both with $d_{model}=2048$), as prescribed by MUP, given our proxy-tuned model was 3x and 8x narrower than these models, respectively ($d_{model} = 256$).  So, in Figure 11, the actual LRs are 8x lower for the bigger models.  The MUP scaling of the LR is noted in the second paragraph of Section 2.2, and details for our proxy model are noted in Appendix A, but we will revise the main paper to note that D2Z continues to dominate 10x decay even as the actual *adjusted* LR, $\eta$, decreases substantially.
>
> ### Regarding "this paper needs to show that the benefits of d2z do not completely vanish at large scales"
>
> Continuing from the previous point, we experimented with a variety of other model widths, up to $d_{model} = 2560$ with a 2.7B-parameter model, in Appendix B.10.  Here, the actual adjusted LR is an order of magnitude smaller than the MUP base LR, but the benefit of D2Z over 10x again grows as model scale increases (from 0.51% at 256M, to 0.6% at 590/1.3B, to 1.21% at 2.3B).  We will revise to refer to these larger-scale results in the main paper.  Currently we only provide a brief pointer to "results with different … model scales (Section B.3, B.10)" in the second paragraph of Section 5.3.  Thanks a lot for raising this!
>
> To further address your concerns, note we have also performed new experiments comparing D2Z and 10x decay by training 1.7B models to *80* tokens-per-parameter (i.e., 137B tokens, 4x beyond training-compute optimality).  At the proxy-tuned base LR, we find D2Z achieves a quite significant 3.0% lower relative loss than 10x decay.  We will add these results to Figure 4 and provide further experimental details in the paper.
>
> Note that these 1.7B models trained to 80 TPP are not necessarily "toy" models by today’s standards; for example, we over-train models of this scale for use as proposal models in speculative decoding (to speed up inference).  Moreover, smaller models like Gemma-2B and Phi-1.5 are used in many production settings.  So, finding the right recipes for over-training models at this scale is an important research direction and a valuable supplementary contribution of this paper.
>
> ### Regarding "the impact of the dataset is another thing that is not discussed"
>
> We should have mentioned the conclusions of these tests in the main paper, as currently we only point the reader to "results with different … datasets (Section B.9, B.10)" in Section 5.3.  We will revise the main paper to note that the benefits of D2Z over 10x do continue across different corpora and across different languages.  Results in the main paper use the SlimPajama dataset, while results with NanoGPT use Open Web Text.  Our scaling law experiments use a combination of the Pile and a proprietary Arabic dataset.  In all cases, D2Z is superior to 10x decay under an optimal LR.  That being said, we will also note in our Limitations section that our results may not hold across all datasets, and understanding the interactions between dataset quality and the optimal decay schedule is an avenue of future research.  Again, good point.

---

> > ### Author Response · Authors · 2024-11-15
> > **Response to Reviewer sx5Y (2/2)**
> >
> > ### Regarding the clarity/formality of the methods
> >
> > We quite appreciate the intuition of your note, "with d2z, the last iterations have an almost zero learning rate, yielding no progress and effectively wasting these last steps. This effect is worsened for the cosine schedule, which spends more time at low lr's."  This is not widely known by the community!  And, in fact, this "loss surface" perspective is exactly the intuition that we formalized through the dual view of the LR schedule.  That is, by converting from the LR schedule to the EMA coefficients, we can see that cosine D2Z does place vanishingly small emphasis on the final weight updates, whereas linear D2Z does this to a lesser extent (Figure 2, right-hand-side, note y-axis is log scale).  We will revise the paper to begin with your intuition, and then develop the connection to the EMA coefficients subsequently.
> >
> > We should also stress that by visualizing the EMA coefficients (as opposed to visualizing the LR schedule), we can see the effect of different *peak* LRs.  The fact the same LR decay "shape" can average the weight updates very differently, e.g., for WSD or Constant schedules (Figures 26 and 27), is an important contribution of our work, and helps to reconcile prior studies.  Furthermore, the EMA perspective does not require any assumptions regarding the curvature of the loss surface: as long as you are using AdamW, the weights at the final iteration are a weighted average of the updates across all iterations, with weighting coefficients that can be calculated based on the LR schedule.  This is what Reviewer f1Pq means by saying that our claims, while not being supported by strict proof, are nevertheless "fairly rigorous."
> >
> > ### Regarding Section 3 making "it seem that the variables x_t in the parameter’s update EMA (eq.3) are external variables to the problem"
> >
> > Yes, this is a very good point.  From a forecasting perspective, here the forecast itself affects future values of the variable being forecast.  Of course, this situation arises whenever a forecast can lead to changes in behaviour (e.g., when forecasting a stock price, disease spread, traffic congestion, etc.).  The issue in our situation is that, without a mechanism to account for these interactions, the contribution of earlier x_t values to the output y_t may be underestimated.  However, since this effect will likely be similar across all the LR schedules, we do not view it as a fatal flaw in terms of getting value from comparing EMA coefficients of different schedules.  We will definitely note this in the paper – do let us know if you feel more discussion is warranted here.
> >
> > ### Regarding "what base lr is used to train the 1.7B models?"
> >
> > In Figure 4, we used the proxy-tuned max LR for all models, `1.6e02` (Table 4).  Again, note however that the MUP parameterization prescribes larger adjustments for larger model scales, so the actual LR used for the 1.7B model in this case is 1/8th of the proxy-tuned max LR.  Also, note again that results for a sweep of max LRs for the 1.7B models are presented in Figure 11 (right-hand-side).
> >
> > ### Regarding "having a non-zero value for the last lr would be better"
> >
> > We will revise to discuss this in the main paper.  In fact, we did experiment with a variety of other minimum LRs for our sparse models, and presented those results in Figure 20 of the appendix.  Interestingly, we saw a fairly linear trend: the lower the minimum LR, the better the loss, and D2Z was still quite better than 75x decay.  We noted, "these are encouraging findings in the sense that D2Z can seemingly be used directly on a range of problems, without having to worry about tuning a problem-specific LR decay ratio (e.g., 50x or 100x)."  We will pursue training with additional decay ratios in advance of the camera submission in order to flesh out this plot further.

---

> > > ### Comment · Reviewer_sx5Y · 2024-11-19
> > >
> > > Dear authors,
> > >
> > > Thank you very much for you detailed reply, that clarifies many questions I had.
> > >  > Regarding "A base lr sweep … for the 1.7B model would benefit the paper greatly"
> > >
> > > Thanks for the reply, I indeed overlooked this table. I would advise putting one more "scaling" figure in the main text, adding to fig.4., showing that the findings of the paper scale with model size.
> > >
> > > > the base learning rates for the small-scale models used in this paper are very high
> > >
> > > Thanks, I was indeed confused by the $\tilde{\eta}$. Maybe it would be worthwhile to put the value of the actual learning rate as well in table 1, so as to dispel any confusion, and so that the scaling factor is immediate (in sec 2.2. the formula is given but not its instantiation for these specific model sizes).
> > >
> > > > this paper needs to show that the benefits of d2z do not completely vanish at large scales
> > >
> > > Thanks for your reply and for running these extra experiments, I think they consolidate nicely the paper.
> > >
> > > > the impact of the dataset is another thing that is not discussed
> > >
> > > "Results in the main paper use the SlimPajama dataset, while results with NanoGPT use Open Web Text." I think it is an important message, as it helps showing the robustness of the paper's analysis.
> > >
> > >
> > > All of these replies alleviate my concerns regarding the applicability of this method to larger scales / different datasets. I thank the authors for these thorough responses.
> > >
> > > Since the empirical validation is the bulk of the paper, I raise my score from 3 to 6.
> > >
> > > However, I still have some concerns regarding the theory presented in this paper.
> > >
> > > For instance, the authors say that  "as long as you are using AdamW, the weights at the final iteration are a weighted average of the updates across all iterations, with weighting coefficients that can be calculated based on the LR schedule". I believe this is in fact true of any reasonable iterative algorithm one would use to train a neural network. My concern is that the update direction at each step of adam strongly depends on the previous iterates. Hence, I cannot see, formally, how looking only at the weighting coefficients can help understand the dynamics.
> > >
> > > In their reply, the authors state that "since this effect will likely be similar across all the LR schedules", but I see neither a proof or an empirical validation of this fact, yet it seems to me that this is the premise upon which all of the theoretical discussion is built upon. It is also a very counter-intuitive fact, since the adam iterates depend on the model's training stage, which itself depends on the decay of the learning rate. Hence, I do feel that more discussion is warranted here.

---

> > > > ### Author Response · Authors · 2024-11-30
> > > >
> > > > Thank you so much for engaging in the discussion period and being so open-minded regarding the paper!
> > > >
> > > > ### Regarding: "However, I still have some concerns regarding the theory presented in this paper."
> > > >
> > > > We will note the following in Section 2.3, when introducing the EMA perspective on AdamW:
> > > >
> > > > > Note the EMA perspective does not account for how weight updates at step $t$, $x_t$, themselves depend on the parameters $y_t$; i.e., the updates and parameters influence each other beyond the explicit connection made by the EMA.  So, while the EMA perspective has formal limitations, we find it useful as part of our *conceptual* model of training (Section 3), in that it predicts behavior that is supported by experiments."
> > > >
> > > > If at all possible, we would really value any feedback you can provide on the proposed revision of Section 3, posted as comment above.  Here we use "model" in the sense of "[all models are wrong, but some are useful](https://en.wikipedia.org/wiki/All_models_are_wrong)." :)

---

### Public Comment · ~Laurence_Aitchison1 · 2024-11-19
**Request to chat.**

This is awesome work!  And thanks for the shout-out to the EMA interpretation of AdamW.  I was wondering if you'd like to chat?  You can look up my details in Wang and Aitchison :-).

(I am not a reviewer, AC etc. on this manuscript).

---

### Author Response · Authors · 2024-11-30
**Improvements to Section 3 (1/2)**

We thank all of the reviewers for their very helpful comments and their support for the paper's core contribution. The paper presents a large-scale, hypothesis-driven study of learning rate schedules. Reviewers all noted the experiments were extensive, effective, and convincing, and they found the paper well-written. Reviewer sx5Y initially had some concerns whether the results would generalize to larger models and datasets, but concerns were alleviated via discussion and additional large-scale experiments.

The remaining concerns center on the methods and writing in Section 3. Reviewer f1Pq urges "more explicit arguments" to improve the readability, while Reviewer 3io8 would like to see stronger theoretical foundations for the success of linear decay-to-zero. Reviewer sx5Y has concerns regarding the formal correctness of the EMA perspective from Wang & Aitchison.

These concerns ultimately all arise from suboptimal organization and writing in Section 3. We failed to make clear that the bias/variance view and extended EMA perspective serve as our *conceptual model* of LLM training, rather than as a formal mathematical theory. The difference becomes subtle because our conceptual model does use mathematics to help predict training dynamics, but it does not let us, for example, formally derive the optimal LR schedule, as has been done in related work under various assumptions. We rely on experiment to fill in theoretical gaps. The value of any conceptual model is whether it makes useful predictions, but we failed to connect our conceptual model to the specific hypotheses that we tested.

We provide here a revised version of Section 3 that addresses these issues. We greatly thank all the reviewers for helping to strengthen the paper.

___
___

# 3: Methods, Conceptual Foundations, and Hypotheses

We now present an extended EMA perspective on AdamW that accounts for time-varying LRs. We then introduce our conceptual model of LLM training, and connect it to the extended EMA perspective. Finally, we outline the specific testable hypotheses that follow from our model.

## 3.1: AdamW as convex combination of weight updates, driven by LR schedule

*largely as in submitted paper, minus second paragraph*

## 3.2: Conceptual model: bias and variance in LLM training

Following [Andriushchenko et al. (2023, Section 4.1)](https://arxiv.org/abs/2310.04415), our main conceptual premise is that in LLM training, there is an initial training phase focused on movement from initial conditions (bias reduction), followed by a later phase bottlenecked by gradient variance. Furthermore, analogous to prior work optimizing the convex loss gap via SGD (Eq. 1), we argue the primary beneficial mechanism of LR decay in LLM training is to reduce gradient variance during later stages of training.

Per-step gradient variance is known to increase over the course of training ([McCandlish et al., 2018](https://arxiv.org/abs/1812.06162)). Very recently, [Zhang et al. (2024)](https://arxiv.org/abs/2410.21676) showed that for an LLM of a fixed size, training with larger datasets corresponds to larger *critical batch sizes*, which directly relates to larger (aggregate) gradient variance via the gradient noise scale ([McCandlish et al., 2018](https://arxiv.org/abs/1812.06162)).

In the EMA perspective, parameters at step $t$ are a convex combination of prior weight updates. The greater the update variance at each step, the greater the number of updates that should be combined in order to reduce overall variance. While the principle is the same, note here variance is reduced by combining updates *across* steps, rather than by increasing batch size at a *specific* step. Now, with a constant LR, update coefficient $c_{t, i}$ decreases exponentially in $t - i$. However, with a decaying LR, scaling down later coefficients $c_{t, i}$ via a lower $\alpha_i = \eta_i\lambda$ corresponds to scaling *up* all $c_{t, k}$ where $k < i$ (cf. Eq. 5). This has the effect of flattening the coefficients, effectively averaging over more outputs (see appendix Figure 26 for a visual contrast between *Constant* and D2Z coefficients, on a log scale). The more we decay, the more updates we average over, and the more variance is reduced.

---

> ### Author Response · Authors · 2024-11-30
> **Improvements to Section 3 (2/2)**
>
> ## 3.2: Conceptual model: bias and variance in LLM training (**continued**)
>
> Higher decay is preferable to simply reducing the *peak* LR because we also need to reduce *bias*, i.e., minimize the contribution of the initial random weights. In the EMA perspective, the contribution of the initial weights to parameters at step $t$ is $c_{t,1} = \prod_{j=2}^{t} (1 - \alpha_j)$. For a constant LR, $c_{t,1} = (1-\alpha)^{t-1}$. For a decaying LR, and where $\alpha_j \ll 1$, coefficient $c_{t,1}$ can be approximated as $c_{t,1} = (1-\bar{\alpha})^{t-1}$, where $\bar{\alpha}$ is the average $\alpha_j$ over the schedule (see Appendix A for a derivation). Exactly as in Eq. 1, bias therefore decreases exponentially in the *absolute* number of steps, $t$, with a rate of decrease that depends on the LR. Crucially, this means that as we train for more total steps (i.e., a higher tokens-per-parameter; TPP), there is a decrease in the *fraction* of steps required for $c_{t,1}$ to become negligible. At higher TPP, bias reduction becomes relatively less important than variance reduction.
>
> ## 3.3: Experimental Hypotheses
>
> ### **Hypothesis 1**: As TPP increases, the relative benefit of D2Z over 10$\times$ decay will increase.
>
> This hypothesis follows from the premise that gradient variance plays a larger role at higher TPP, and greater LR decay (as in D2Z) allows for more updates to be averaged, and thus greater variance reduction. A related hypothesis is that if we increase the batch size at each step, gradient variance will decrease, and so the benefits of D2Z over 10$\times$ decay should diminish. Note our conceptual framework does not say precisely at which TPP or batch size D2Z will first prevail; here we will rely on our empirical findings (Section 4) to fill in the theoretical gap.
>
> ### **Hypothesis 2**: As TPP increases, the optimal peak LR will decrease, for all LR schedules.
>
> Tuning the peak LR is about trading off movement from initial conditions (requiring a high LR) and mitigating variance (requiring a low LR). As TPP increases, and bias reduction plays a smaller role, we should observe the optimal peak LR to decrease. We hypothesize the decrease will be greater with a *Constant* schedule, as *Constant* does not use decay to balance the conflicting demands of bias and variance. Moreover, the optimal peak LR for other continuous LR schedules, like *WSD* and *Cyclic*, should also decrease with longer training durations. In this way, such "schedule-free" approaches are *not truly schedule-free*. This dependence is obvious when plotting the EMA coefficients for these schedules, as in appendix Figure 27.
>
> This hypothesis also implies that when comparing LR schedules, the maximum LR could be a confounder. For this reason, in our experiments, we compare schedules when each is tuned to their optimal maximum LR.
>
> ### **Hypothesis 3**: *Linear* D2Z will improve over *Cosine* D2Z.
>
> While LR decay allows averaging over more weight updates, if the LR decays too quickly, the final weight updates may not contribute to the EMA. From a loss surface perspective, as the LR approaches zero, we take vanishingly small steps. Since *Cosine* reaches smaller step sizes faster than *Linear* (left side, Figure 2, Section 1), it will make less progress toward the optimum loss. From the EMA perspective, this is equivalent to *Cosine* having smaller $c_{t,i}$ coefficients as $i$ approaches $t$ (Figure 2, right). Note this problem is unique to *Cosine* D2Z and will not affect, e.g., *Cosine* 10$\times$ decay.
>
> ### **Hypothesis 4**: A high LR (not weight decay), is needed to reduce bias and achieve optimal loss.
>
> Note that weight updates have a coefficient of $\frac{1}{\lambda}$ in Eq. 4. So, while $\eta_t$ and $\lambda$ contribute equally to $\alpha_j$, increasing $\lambda$ to reduce bias is counterproductive as weight updates will be scaled down proportionally, reducing movement from initial conditions. However, if LR is in a high enough range, both LR and WD should equally affect variance reduction.
>
> Since weight decay does not impact bias reduction, then at very high TPP, it should be more effective to reduce variance by lowering WD than by reducing the LR. This stands in contrast to very recent work that recommends purely decreases in LR for high-TPP training ([Bjorck et al., 2024](https://arxiv.org/abs/2409.19913)).

---

### Meta-Review · Area_Chair_9rzG · 2024-12-10

**Metareview:**

The paper studies the impact of learning rate scheduling on LLM training, focusing on a linear decay to zero schedule. It empirically demonstrates that this schedule outperforms cosine decay schedules in terms of convergence and robustness across varying hyperparameters. The work also introduces a perspective of AdamW as an exponential moving Average (EMA) perspective.


### **Strengths**
- Tackles an important practical problem
- Extensive experiments across various models and datasets demonstrate the interest in the specified learning rate schedule

### **Weaknesses**
- No theory or strong justification for the schedule
- The empirical gains are quite small (see e.g. Table 1)

Overall, the paper reports interesting empirical findings and I therefore lean toward acceptance. I encourage the authors to enhance the discussion regarding the shortcomings of the paper.

**Additional Comments On Reviewer Discussion:**

Some reviewers increase their scores after the discussion period.

---

### Decision · Program_Chairs · 2025-01-22

Accept (Poster)